# Rapid vessel segmentation and reconstruction of head and neck angiograms using 3D convolutional neural network

Fan Fu [1,2], Jianyong Wei[3], Miao Zhang[1,2], Fan Yu[1,2], Yueting Xiao[3], Dongdong Rong[1,2], Yi Shan [1,2], Yan Li[1,2], Cheng Zhao[1,2], Fangzhou Liao[4], Zhenghan Yang[5], Yuehua Li[6], Yingmin Chen[7], Ximing Wang[8] & Jie Lu [1,2,9 ✉]

The computed tomography angiography (CTA) postprocessing manually recognized by technologists is extremely labor intensive and error prone. We propose an artificial intelligence reconstruction system supported by an optimized physiological anatomical-based 3D convolutional neural network that can automatically achieve CTA reconstruction in healthcare services. This system is trained and tested with 18,766 head and neck CTA scans from 5 tertiary hospitals in China collected between June 2017 and November 2018. The overall reconstruction accuracy of the independent testing dataset is 0.931. It is clinically applicable due to its consistency with manually processed images, which achieves a qualification rate of 92.1%. This system reduces the time consumed from 14.22 ± 3.64 min to 4.94 ± 0.36 min, the number of clicks from 115.87 ± 25.9 to 4 and the labor force from 3 to 1 technologist after five months application. Thus, the system facilitates clinical workflows and provides an opportunity for clinical technologists to improve humanistic patient care.

[1] Department of Radiology, Xuanwu Hospital, Capital Medical University, No. 45 Changchun Street, Xicheng District, 100053 Beijing, China. [2] Beijing Key Laboratory of Magnetic Resonance Imaging and Brain Informatics, No. 45 Changchun Street, Xicheng District, 100053 Beijing, China. [3] Shukun (Beijing) Technology Co., Ltd., Jinhui Building, Qiyang Road, 100102 Beijing, China. [4] Institute of Information Engineering, Chinese Academy of Sciences, No. 52 Minzhuang Road, 100093 Beijing, China. [5] Department of Radiology, Friendship Hospital, Capital Medical University, No. 95 Yongan Road, Dongcheng District, 100050 Beijing, China. [6] Institute of Diagnostic and Interventional Radiology, Shanghai Jiao Tong University Affiliated Sixth People's Hospital, No. 600, Yi Shan Road, 200233 Shanghai, China. [7] Medical Imaging Department, Hebei General Hospital, No. 348 Hepingxi Street, 050051 Shijiazhuang, Hebei, China. [8] Department of Radiology, Shandong Provincial Hospital, No. 324 Jingwei Road, 250021 Jinan, Shandong, China. [9] Department of Nuclear Medicine, Xuanwu Hospital, Capital Medical University, No. 45 Changchun Street, Xicheng District, 100053 Beijing, China. ✉email: imaginglu@hotmail.com

Computed tomography angiography (CTA) is a widespread, minimally invasive, and cost-efficient imaging modality that is employed in routine clinical diagnoses of head and neck vessels[1], especially in cerebrovascular disease, which represents one of the leading causes of severe disability and mortality worldwide[2]. To visually analyze the vasculature of the head and neck more efficiently, image reconstruction is usually performed by experienced computed tomography technologists. However, as the number of requests for CTA examinations have increased, the postprocessing staff cohort has become overwhelmed due to the time-consuming manual process[3]. Considering that vessel imaging reconstruction is required in clinical settings, an automatic reconstruction system can be easily integrated into the clinical workflow if processed segmentations are available.

Deep learning-based segmentation approaches have drawn increasing interest due to their self-learning and generalization abilities from large data volumes[4]. A major technological challenge of this study is accurate vessel segmentation without interruption of the blood vessels which is difficult because the vessels are tortuous and branched. Although most rule-based approaches such as centerline tracking, active contour models, or region growth exploit various vessel image characteristics to reconstruct vessels[5,6], they are either hand-crafted or insufficiently validated[7,8]; thus, achieving the desired level of robustness on full brain vessel segmentation is difficult. In addition, segmentation is easily affected by other tissues; for example, the CT value of the blood vessel at the intracranial entrance is highly similar to the CT value of the cranium, which causes extravascular tissue adhesion and interferes with subsequent disease diagnosis. Therefore, accurate segmentation extraction of arterial vessel status is not available; instead, a system with adaptive properties that is capable of automatic segmentation with continuity is required.

Deep neural network architectures are the most effective way to overcome the above technological roadblocks[9]. In the past few years, various deep learning models have shown significant potential for image classification and segmentation tasks in various fields, particularly in neuroimaging[10]. Specifically, U-net, which is a specialized convolutional neural network (CNN), has demonstrated its promise in medical image analysis[11]. Moreover, a 3D-CNN model that conforms to the physiological, anatomical, and morphological features of the objective images is specifically designed for segmentation tasks and has shown high segmentation performance for biomedical images[12].

In this present study, we sought to develop an automatic imaging reconstruction system (CerebralDoc) based on an optimized anatomy prior-knowledge based 3D-CNN to reconstruct original head and neck CTA images, assist technologists in their daily work and establish a time-saving work process. A thorough quantitative assessment is performed from three perspectives to evaluate the strength of CerebralDoc, including the algorithmic performance of the model, whether the image reconstruction quality satisfies diagnostic needs, and the efficiency of its clinical application with model augmentation. Collectively, we argue that artificial intelligence (AI) technology can be integrated into the radiology workflow to improve workflow efficiency and reduce medical costs.

## Results

**Patients and image characteristics.** A total of 18,766 patients consecutively subjected to head and neck CTA scans at five tertiary hospitals were included in this study. Due to poor image quality, 507 scans were excluded via manual inspection. The mean age of the participants was $63 \pm 12$ years: 9370 patients (51.3%) were male, and 8889 patients (48.7%) were female. The patient cohort characteristics and those of the head and neck CTA scans that were employed for training, validation, and independent testing data sets are summarized in Table 1. The overall experimental design of the workflow diagram is shown in Fig. 1.

**Model performance.** A total of 91,295 augmented images were employed for training. The 3D-CNN model was trained using a modified U-net with the addition of bottleneck-ResNet (BR) which automatically achieved optimized model parameter selection. The core design of the CerebralDoc system was the deep learning model, which was divided into two parts: ResU-Net and connected growth prediction model (CGPM). ResU-Net was primarily responsible for bone segmentation and vessel extraction, while the proposed CGPM was utilized to ensure vessel integrity. In part one, for feature map extraction, we applied the

**Table 1 Basic characteristics of enrolled subjects, number of CT images from different manufacturers, and number of different diseases.**

| Parameters | Patient (images) metric |
|---|---|
| Patients characteristics | |
| Number of patients (training/independent testing/clinical evaluation and application data sets) | 16,433/1826/152/3430 |
| Male to female (training/independent testing/clinical evaluation and application data sets) | 1.05:1 (6539:6242)/1.07:1 (2831:2647)/1.17:1 (82:70)/1.03:1 (1738:1692) |
| Age (y) (training/independent testing/clinical evaluation and application data sets) | $64 \pm 11/62 \pm 14/66 \pm 13/64 \pm 13$ |
| Different manufactures | Number of patients in training/testing data sets |
| GE | 7071/1011 |
| Siemens | 5271/523 |
| Philip | 2556/197 |
| Toshiba | 1535/95 |
| Different diseases | Number of patients in training/testing/clinical evaluation/ application data sets |
| Atherosclerosis | 11,503/1223/128/2982 |
| Cerebrovascular disease | 5752/548/52/1744 |
| Arterial aneurysm | 679/70/7/176 |
| Moyamoya disease | 159/17/3/45 |
| Interventional therapy | 448/46/2/138 |
| Vascular variation | 247/26/2/59 |

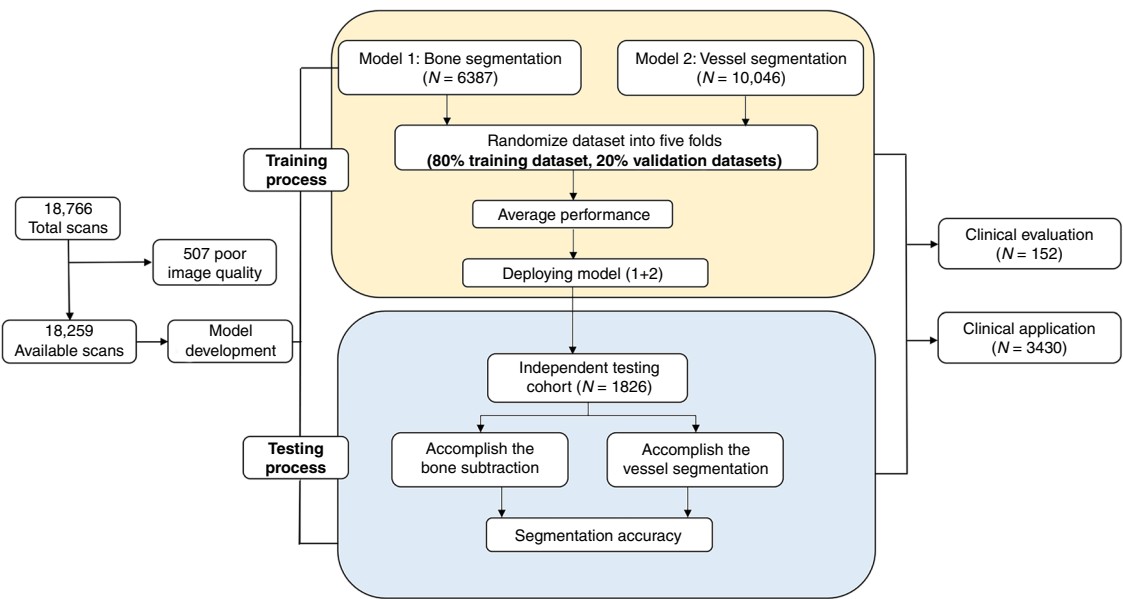

**Fig. 1 Data flow diagram showing our approach to achieve bone and vessel segmentations.** The training process was divided into two parts with two separate training cohorts. Two models derived from ResU-net were accessed for performance evaluation by fivefold cross-validation and were subsequently merged to form the first layer of CerebralDoc. The validation of bone and vessel segmentation which against to the annotated source images was used to make sure the algorithm indicators acceptable. We deployed the final two models to an independent cohort that contained 1826 cases, including almost all major CTA platforms on the market, to manifest the credibility of CerebralDoc by accuracy analysis.

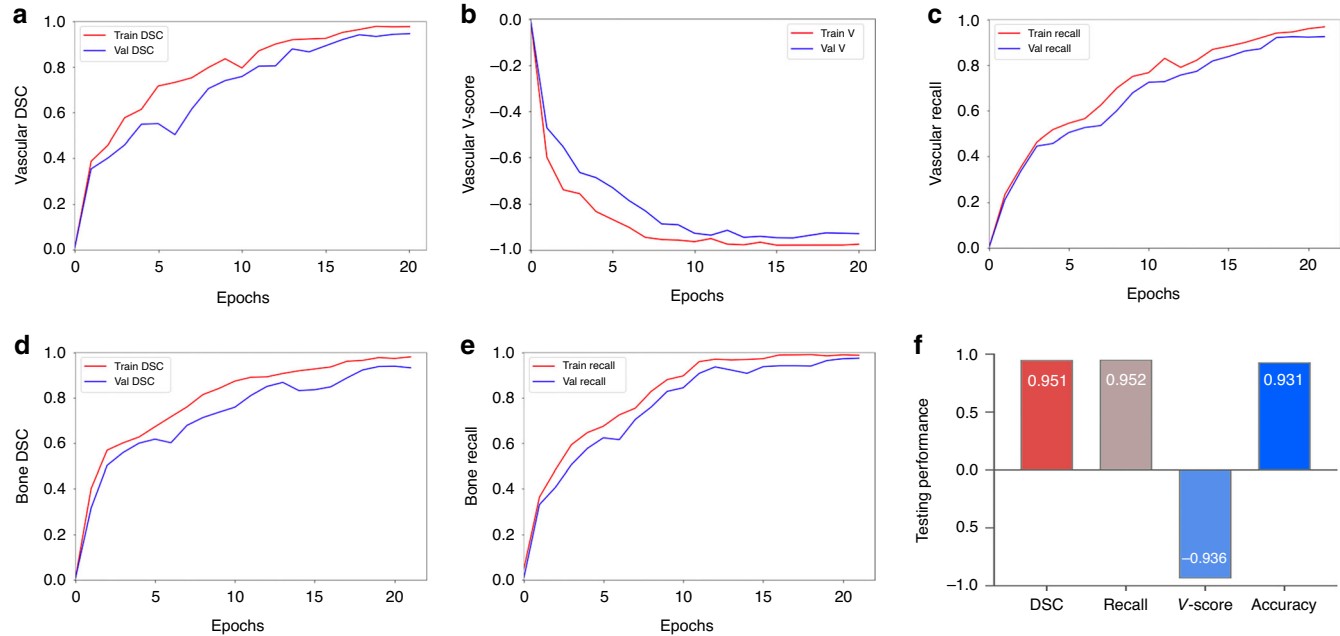

**Fig. 2 The performance of the training and testing process of head and neck CTA postprocessing.** Illustration of the changes of the DSC value (**a**), $V$-score value (**b**), and the recall value (**c**) over the training and validation sets in the vessel segmentation task. **d**, **e** The variety of the DSC value and recall value over the training and validation sets in the bone segmentation task. The overall DSC, $V$-score, recall, and accuracy values in testing set of the pipeline are shown in **f**.

distribution of BRs from the ResU-Net base, which exhibited the optimal segmentation performance of the model. The model training process included both bone segmentation and vessel segmentation, and all the curves reached convergence by the 20th epoch. The vessel segmentation attained dice similarity coefficient (DSC) of 0.975 and 0.944, weighted vessel score ($V$-scores) of −0.975 and −0.929, and recall of 0.961 and 0.933 for the training and validation sets, respectively. The bone segmentation obtained DSC of 0.979 and 0.960, recall of 0.984 and 0.976 for the training and validation sets, respectively. Figure 2 shows a detailed image demonstrating the evolution of the DSC, loss function value, and the segmentation recall over the training and validation sets. In the testing process, the model was improved with a DSC of 0.951, $V$-scores of −0.936, and recall of 0.952, compared with the

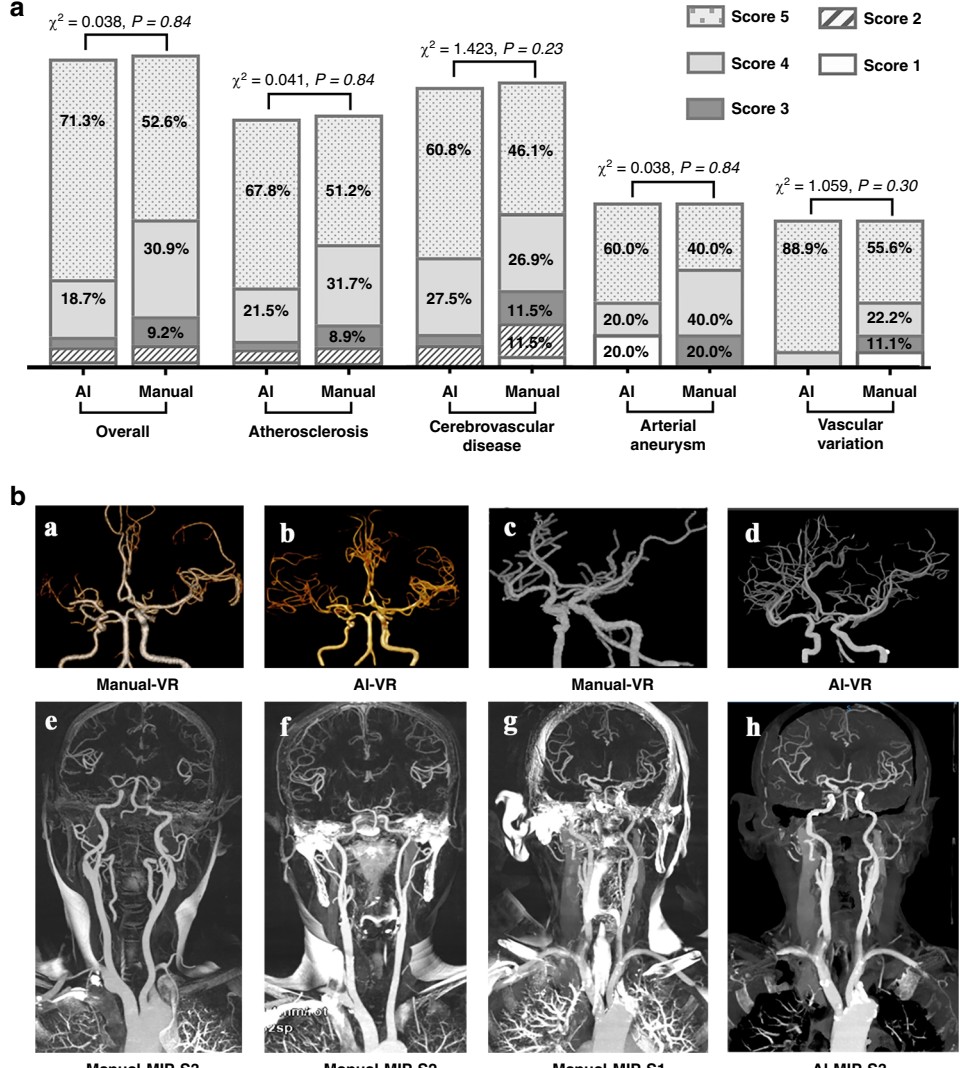

**Fig. 3 Comparison of the image quality of the head and neck CTA between CerebralDoc and manual processing. a** distribution of the 5-points rating in the overall clinical evaluation data set and different diseases. A nonparametric test was used for comparing two groups. **b** Comparison between AI and manual processing in VR and MIP. a Right middle cerebral artery is occlusive without reconstruction of collateral circulation in manual processing, while AI successfully reconstructed the collateral circulation (b). The VR was cleaner in AI than the manually processed image, especially in cavernous sinus segments (c, d). e–g The standard for evaluating the MIP in the manually processed image and h was generated by CerebralDoc for the same patient as g without bone residual.

original model (DSC of 0.925, $V$-scores of −0.916 and recall of 0.933). The overall accuracy of segmentation is 93.1%. 32 (1.75%) failed bone subtraction and 84 (5.15%) failed vessel segmentation.

**Clinical evaluation of CerebralDoc.** We assessed the model in terms of its ability to automatically achieve head and neck CTA reconstruction, evaluated whether the reconstructed image quality satisfies the radiologist's diagnostic needs, and compared the performance to human output. The concordance between the two raters on the independent testing data set of CerebralDoc and human output was satisfactory (Fleiss' $\kappa = 0.91$ and 0.88). Among the 152 scans, the overall qualification rates of CerebralDoc and human output were 92.1% (140/152) and 93.4% (142/152), respectively, according to whether the reconstructed images were sufficient for rendering clinical diagnosis possible. However, the excellent image quality score of 5 for CerebralDoc was higher than that for human output [70.4% (107/152) vs 52.6% (80/152), $\chi^2 = 11.199$, $P = 0.01$]. In the evaluation of the qualification rate

for different diseases, including atherosclerosis, cerebrovascular disease, arterial aneurysm, and vascular variation, there was no statistically significant difference between CerebralDoc and human output (Figs. 3a and 4).

In a qualitative evaluation of volume rendering (VR), maximum intensity projection (MIP), curve planar reconstruction (CPR), and curved multiple planar reformation (MPR), there were statistical differences with respect to VR ($P < 0.001$) and MIP ($P < 0.001$) in favor of CerebralDoc (Supplementary Table 1). However, there were no significant differences in CPR and curved-MPR between the two outputs. The improved visualization of VR was the farther cerebral vascular branches and clearer vessel image, especially in the evaluation of compensatory circulation of the ischemic region (Fig. 3b (a, b)) and cavernous sinus segments [Fig. 3b (c, d)]. In the qualitative evaluation of MIP generated by manual processing, 73 (48.0%) were rated with a score of 3, 44 (28.9%) were rated as having moderate bone residue but do not affect vessel observation with a score of 2 and 35 (23.1%) were rated as having severe bone residue and do affect

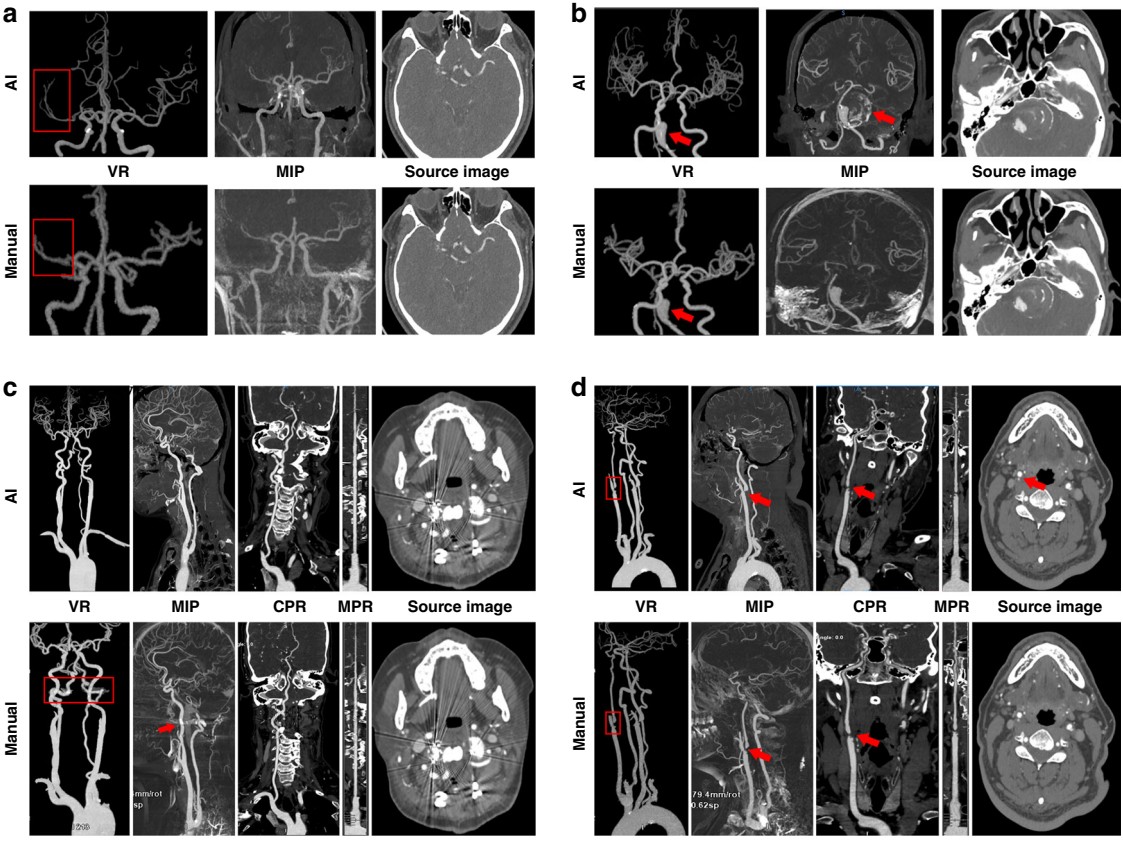

**Fig. 4 The direct presentation of images generated by both CerebralDoc and manual processing. a** The right middle cerebral artery is occlusive without the establishment of collateral circulation. **b** Basilar artery aneurysm with thrombosis and calcification, which can be observed in the MIP reconstructed by CerebralDoc. **c** Metal artifact was suppressed better in AI after the Atlanto-occipital surgery. **d** Severe stenosis of bifurcation of the right common carotid artery and left vertebral artery arising directly from the aorta. Both AI and manual outputs were excellent.

vessel observation [Fig. 3b (e–g)]. In CerebralDoc, all the MIP images were rated with a score of 3 due to automatic bone segmentation in the algorithm. Moreover, MIP generated by AI directly could observe the wall calcification (Fig. 3b (h)).

For the disqualified cases, 10 (6.6%) cases were rated as having weak or poor image quality and 2 cases failed to reconstruct in CerebralDoc. Among them, partial vessel interruption occurred in 6 cases (3.9%), veins were misidentified as arteries in 2 cases (1.3%) and curved-MPR was lacking in 2 cases (1.3%). A possible reason for the 2 failed cases was vascular abnormal routine and weak CT signals. Among the 10 failed cases of human output, the severe bone residual was detected in 6 cases, vein artifacts were observed in 3 cases and multiple vessel interruption occurred in 1 case (Fig. 5).

**Clinical application efficiency of CerebralDoc.** CerebralDoc was applied to all the patients who underwent head and neck CTA examinations, including patients with atherosclerotic ulceration, aneurysm, moyamoya disease, and cervical stent implantations (Fig. 6). From July to November 2019, a total of 3430 patients underwent head and neck CTA scans at Xuanwu Hospital in the clinical setting; of these, 3122 (91.0%) were successfully reconstructed by the software as evaluated by the experienced technologists. Finally, 2649 reconstructed images were applied in the clinical context (Fig. 7a). Moreover, as the doctors' trust improved, the probability of the reconstructed images being pushed to the clinic gradually increased from 243 to 753 (Fig. 7b).

**Comparing the performance of CerebralDoc with the manual process.** The average postprocessing time of two technologists

was 13.48 ± 3.67 and 14.95 ± 3.65 min. Compared with CerebralDoc, the average time consumed was reduced from 14.22 ± 3.64 to 4.94 ± 0.36 min ($P < 0.001$, Fig. 8a). We also found that the consumed postprocessing time and click numbers were different in patient and normal person by manually recognized ($P < 0.001$), while there was no significant difference in AI (Fig. 8b, d). Moreover, the number of clicks was reduced from 115.87 ± 25.9 to 4 ($P < 0.001$), and the labor saved reduced the necessary labor force from 3 technologists to 1 (Fig. 8c). In addition, the number of CTA examinations doubled during the five months during which CerebralDoc was applied, and the success rate of the software gradually increased.

**Discussion**

In this study, we innovatively developed and validated a deep learning algorithm through physiological anatomical-based 3D-CNN for CTA head and neck image postprocessing. The model was developed with an optimized network by the distribution of BRs, and we proposed the CGPM to revise vascular segmentation errors and avoid partially missing vasculature. Moreover, we applied CerebralDoc to clinical scenarios for five months in Xuanwu Hospital to use an innovative and optimized deep learning algorithm that uses artificial intelligence to reconstruct CTA images used directly for head and neck vascular disease diagnoses. Our results indicated that a 3D-CNN deep learning algorithm can be trained to complete bone and vessel segmentation automatically with high sensitivity and specificity in a wide variety of enhanced CTA scans, including the aorta, carotid artery, and intracranial artery. Strikingly, CerebralDoc's reconstructed image quality performance was evaluated by two

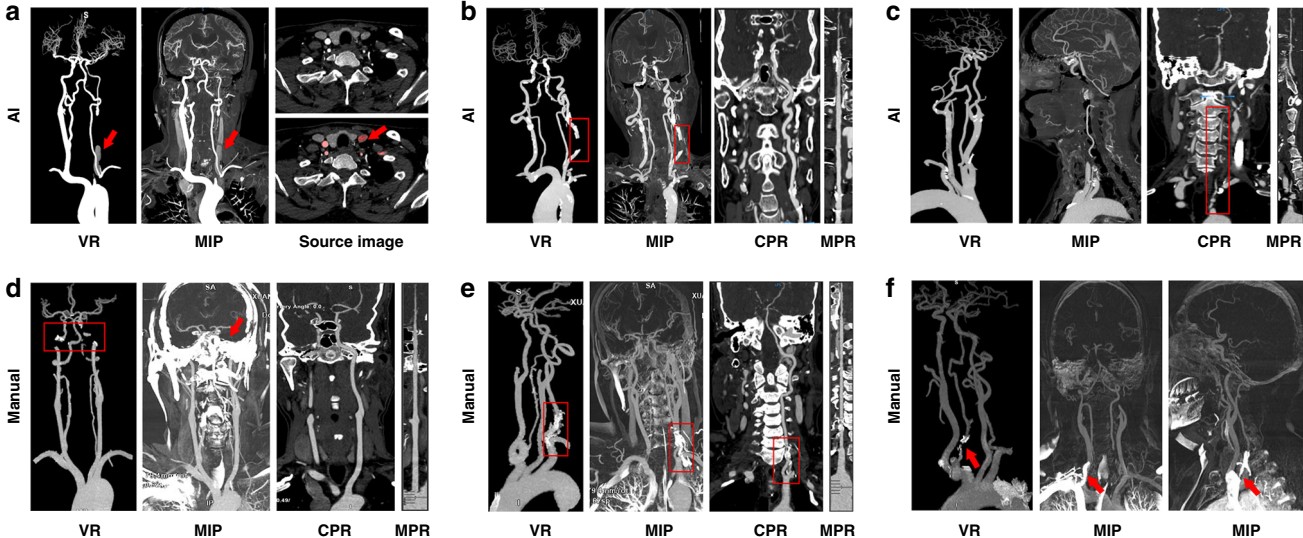

**Fig. 5 Some erroneous postprocessing of CerebralDoc and human outputs. a** Misidentification of veins as arteries. **b** Partial vessel interruption. **c** Left vertebral artery arising directly from the aorta; the CPR and curved-MPR failed. **d** Severe bone residual (red arrow) caused the vessel interruption in VR and MIP. **e** Bone remains at the origin of the left vertebral artery. **f** Vein artifact at the origin of the right vertebral artery.

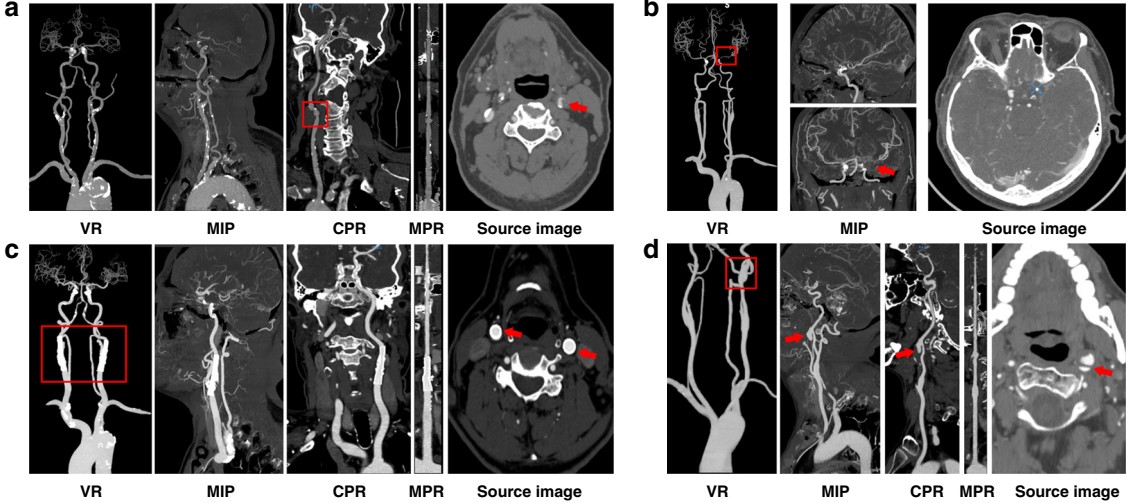

**Fig. 6 Several accurate postprocessing of CerebralDoc in the clinical application. a** Atherosclerotic ulceration in the bifurcation of the left common carotid artery. **b** Moyamoya disease in the left internal carotid artery. **c** Cervical stent implantations in the bilateral internal carotid artery. **d** Aneurysm in the left internal carotid artery.

radiologists with significant clinical experience; the performance showed a qualification rate of 92.1%. Therefore, this creative pipeline for the reconstruction of head and neck CTA offers numerous advantages, including consistency of interpretation, labor and time savings, reduced radiation doses, and high sensitivity and specificity.

Prior studies that have used deep learning for vessel segmentation tasks have mostly focused on algorithm optimization and model organization, which encouraged our development of a purposed automated clinical vessel segmentation tool (Supplementary Table 2). For example, a Y-Net model was developed to achieve 3D intracranial artery segmentation from 49 cases of magnetic resonance angiography (MRA) data and has achieved a precision of 0.819 and a DSC of 0.828 for the testing set[13]. The performance of this model was shown to perform better than the three traditional segmentation methods in both binary classification and visual evaluation. In addition, Michelle et al. have

proposed HalfU-net to perform the images reconstruction by 2D patches, which yielded high performance for time-of-flight (TOF)-MRA vessel segmentation in patients with cerebrovascular disease: a dice value of 0.88, a 95 Hausdorff distance (HD) of 47 voxels and an absolute volume difference (AVD) of 0.448 voxels[9]. Furthermore, MS-Net, which was proposed to improve segmentation accuracy and precision, and significantly reduces the supervision cost, was proposed for full-resolution segmentation[14]. These previous works demonstrated satisfactory performance in specific vessel areas segmentation, such as aortic[15,16], carotid[17,18], or intracranial vessels[19], which particularly inspired our study. However, complete head and neck CT scans that include three different vessels with different size magnitudes (aorta, carotid arteries, and intracranial blood vessels) make it difficult for models to capture cross-size-grade vessel characteristics. In addition, most previous studies were derived from relatively small data sets that were limited and lacked

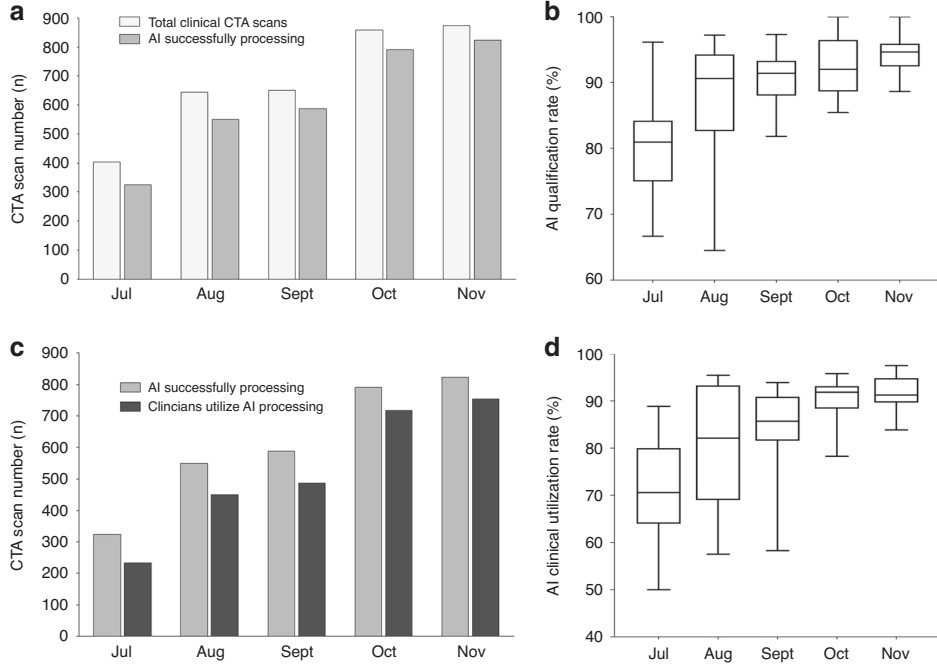

**Fig. 7 The clinical application of CerebralDoc from July to November 2019 ($n = 3430$ patients were included).** The qualification rate = AI successfully processing numbers/total clinical CTA numbers. **a** The number of clinical CTA scans increased from 404 to 874 and the AI qualified number doubled within the five months. **b** The AI qualification rate has a 10.92% improvement. **c** The clinical utilization number increased fourfold. **d** The usage rate ranges from 71.15–91.48%. In **b**, **d** data are represented as boxplots where the middle line is the median, the lower and upper hinges correspond to the first and third quartiles, the upper whisker extends from the hinge to the largest value no further than 1.5 × IQR from the hinge (where IQR is the interquartile range) and the lower whisker extends from the hinge to the smallest value at most 1.5 × IQR of the hinge.

validation from real clinical data. Therefore, a comprehensive pipeline with a guaranteed maximum patch-size ($256 \times 256 \times 256$ voxels) for automatic head and neck CTA postprocessing by optimized physiological anatomical-based 3D-CNN is of great significance in clinical practice.

The application of a modified U-net framework was suggested by Ronneberger et al. in 2015[11], and previous studies have used this strategy on large images cropped into smaller pieces, causing the loss of the structural characteristics of the entire vessel[20–22]. Considering the importance of vessel integrity, our strategy instead inputs the original 3D slices into the 3D CNN with cropping and modifies the U-net architecture with the distribution of BR to automatically find the optimized model parameters. This approach achieves significant parameter reduction and results in high quantitative performance for head and neck vessel segmentation. Additionally, connecting the three different size levels of vessels is still a challenge. Some studies used level sets for multi-size vessel segmentation tasks, but these methods were not strictly automated and required user intervention during the process[23,24]. Based on the physiological anatomical information, we divided the entire volume into three regions (aorta, carotid, and intracranial regions) for the network to achieve better feature extraction of vessels of different sizes. Moreover, we proposed the CGPM by learning from the input data of original CTA images without labeling, the current segmentation result generated by ResNet1, ResNet2, and ResNet3, and the labeled CTA images to revise vascular segmentation errors and effectively avoid partial missing vasculature.

Rarely have studies investigated vessel segmentation from the head and neck CTA scans due to the wide range from the aorta to skull and the tortuous and branched arterial brain vessel. Meijs et al.[25] proposed a robust model for CTA cerebral vascular segmentation in suspected stroke patients, but this method used hand-crafted features, which are time-consuming and difficult to obtain. Michelle et al.[9] utilized the U-Net deep learning framework to perform MRA vessel segmentation on 66 patients with cerebrovascular disease; this model served as the starting point for developing a model applicable to clinical settings. Our study is a multicenter, large sample study to complete the automatic framework of bone segmentation and vessel segmentation from the aortic arch to the intracranial artery in accordance with the clinical postprocessing process. We translated this process into clinically applicable software named CerebralDoc, which is available for research and clinical practice.

To provide a referenceable benchmark, we evaluated the performance of CerebralDoc on one prospective database. As expected, its overall qualification rate and performance in different diseases were not significantly different from those in manual processing. In the visualization of pathology, VR and MIP significantly improved CerebralDoc in the presentation of cerebral vascular branches, vessel cleanness, and bone removal compared to manual reconstruction. A previous study revealed the advantage of deep learning in providing a more reliable recognition performance by exploring the 3D structure[26]. It also worked in our study by improving the inspection of vessel cleanness; the 3D-CNN could learn the vessel feature instead of the threshold value to accurately improve and capture the vessel signals[27]. VR also improved the resolution of the distal and small cerebral vessels in CerebralDoc, which benefit from the ability of deep learning to learn the edge characters and morphological features of vessels[28]. Moreover, accurate bone removal is an immense challenge in head and neck CTA reconstruction because patient motion between the plain scan and enhanced scan is inevitable, which severely affects the subtraction result. Thus, the MIP sequence would contain severe bone residuals, which affect the observation of vessels in manual reconstruction. However, this pipeline can serve as an automatic bone removal tool by the proposed ResU-Net, which applied the enhanced images only

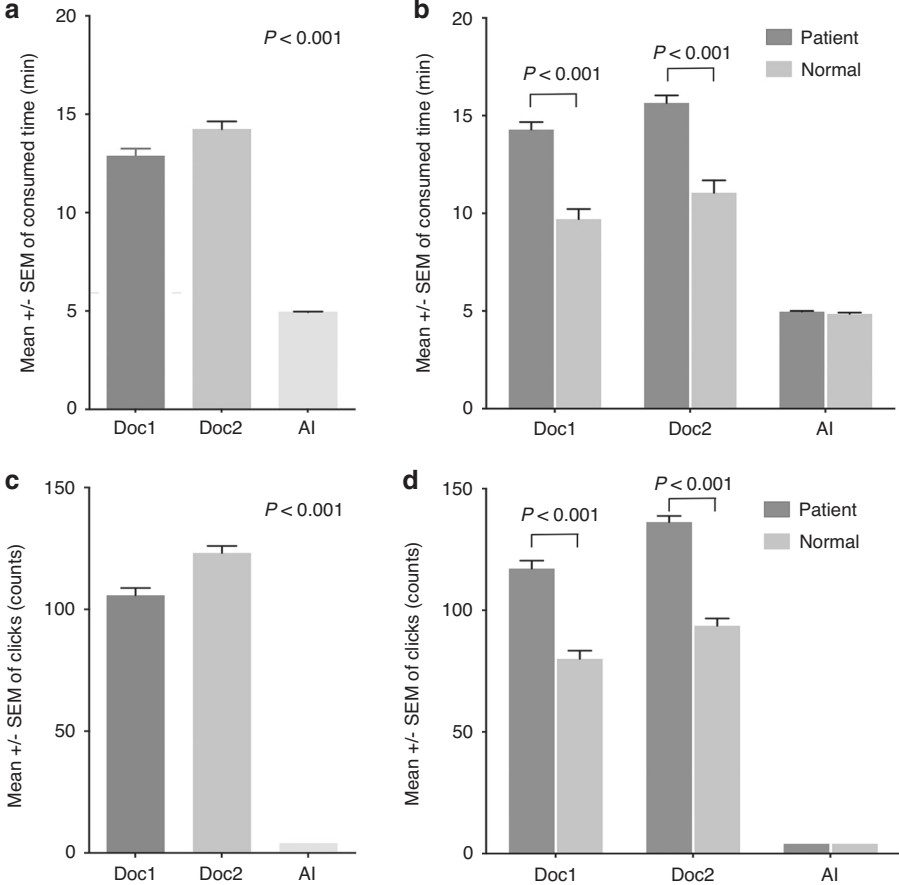

**Fig. 8 The comparison between the CerebralDoc and traditional manual process (*n* = 100 patients were randomly selected). a** The average consumed time of two technologists and AI. **b** The difference of the postprocessing time in patient and normal person between technologists and AI. **c** The number of clicks of two technologists and AI. **d** The difference of the click numbers in patient and normal person between technologists and AI. All data were shown as error bars and expressed as mean values ± SEM. Unpaired Student's *t*-test was used for comparing two groups and Ordinary one-way ANOVA with Dunnett multiple comparison test for multiple comparisons.

instead of the enhanced and plain image subtraction and reduced the radiation dose. This combination of less bone residual and lower radiation dose could be of an important significance for clinical use. Besides, MIP contained the wall calcification in CerebralDoc, which is beneficial for radiologists to evaluate directly and intuitively. The vessel interruption caused by severe stenosis in CerebralDoc was mostly solved at the end of November by collecting these cases and re-training the model. Additionally, the reason for vein misidentification was the delayed contrast phase which was inevitable due to the algorithm limitation.

In this study, we developed a computational tool that can automatically segment and extract entire vessels from head and neck scans. This tool can potentially assist technologists in clinical practice by quickly pinpointing vessels in axial CT images automatically. It is time-consuming and experience-dependent for technologists to locate very small vascular regions; thus, our approach greatly reduces the time that technologists must spend on each image. Second, this tool can help technologists remove artifacts caused by veins or metals in most cases and readily provide complete reconstructed images. Above all, this tool can be used to access head and neck postprocessing CTA images, including CPR, VR, MIP, and curved-MPR, which are useful for assisting radiologists and clinicians in the diagnosis of vascular disease after the quality of the images has been evaluated.

Although the results of this study are promising, it does have some limitations. In this pilot study, images from patients with severely anomalous origins (abnormal vessel routine except for left vertebral arising directly from the aorta and severely stenotic origin) of head and neck arteries were not included in the training and validation sets. Therefore, further clinical collection and testing is needed to assess the clinical accuracy for various forms of skull and neck vessels. Due to the relatively low incidence of such deformities, this problem does not affect our overall conclusion. This study was performed for over 2 years, and CerebralDoc functioned properly for five months. However, further evaluation of the reconstruction tool is needed to assess long-term accuracy and stability needs. In addition, this deep learning algorithm is dependent on big data; therefore, it is not appropriate for small sample sizes. Noisy images also need to be tested in future work to more fully assess the system's robustness. In addition, CerebralDoc currently works only on head and neck CTA images.

In conclusion, CerebralDoc is a practical system to provide head and neck CTA reconstruction. It offers a time-saving and subjectivity-independent method compared to currently available techniques to optimize the reconstruction of CTA images, saving costs, and increasing efficiency. Due to the automatic AI-based standardized U-net and the generation of visualized postprocessed images, CerebralDoc has the potential to become a standard component of the daily workflow, leading to the establishment of a process for radiological operations that can support increased communication with patients allow more humanistic care to be provided in the future.

## Methods

**Data preparation.** The study was approved by the institutional review board of the University Medical Center. Institutional Review Board (IRB)/Ethics Committee approvals and informed consents were obtained in Capital Medical University Xuanwu Hospital. The data set used in this study consisted of 18,766 raw contrast-enhanced head and neck CTA images collected retrospectively and randomly from five tertiary hospitals in China, including 8593 scans from Xuanwu Hospital of Capital Medical University, 4407 from Beijing Friendship Hospital affiliated to Capital Medical University, 2584 from Shanghai Sixth People's Hospital, 2096 from Hebei General Hospital and 1086 from Shandong Provincial Hospital. All the head and neck CTA scans were performed as part of the patients' routine clinical care. Each scan included the aorta, carotid artery, and intracranial artery with ~561–967 slices in the $Z$ axis. We excluded 507 (2.7%) CTA scans based on poor imaging quality or severe artifacts via manual inspection. All the patient CTA images were in DICOM format, and the scans were acquired from GE, Siemens, Philip, and Toshiba equipment using 64-, 192-, 256- or 320-detector row CT scanners (GE Revolution, GE Healthcare, Boston, USA; Somatom Force, Siemens Medical Solutions, Forchheim, Germany; Somatom Definition Flash, Siemens Medical Solutions, Forchheim, Germany; Philips Brilliance ICT 256-slice, Philips Medical Systems, Best, the Netherlands; Toshiba Aquilion one, Toshiba Medical Systems, Tokyo, Japan). Data were acquired by using 64, 192, 256, or $320 \times 0.5$, 0.625, or 0.7 mm collimations and a rotation time between 0.5 and 0.7. The primary thickness ranged from 0.4 to 0.6 mm; and the contrast agent was iohexol (Guerbet, France) at a concentration of 300 mgI/ml and an injection rate of 5 ml/s.

In this study, we initially obtained 18,259 patients with 14,461,128 head and neck CTA scanning images from January 2018 to February 2019 acquired mainly from four different CT manufacturers. During the training process, scans of 16,433 patients were divided into two models. Model 1 consisted of scans from 6387 patients for bone segmentation and model 2 consisted of scans from 10046 patients for vessel segmentation. During the testing process, scans from another 1826 patients were assigned to an independent testing cohort and used for performance assessment.

For the clinical evaluation set, 152 additional head and neck CTA scanning images were prospectively selected from Xuanwu Hospital between May 2019 and June 2019 to evaluate the clinical performance of the model. These images were also reconstructed by the technologists as a comparable output. Two experienced radiologists were blinded to the imaging sources and assessed whether the imaging quality fulfilled the diagnostic requirements based on a 5-point scale randomly.

Finally, a clinical application set was collected mainly to verify the performance of the proposed model in real clinical application scenarios from July to November 2019, this set, included the number of postprocessing operations, the time saved, and the clinical productivity.

**Radiologist annotations.** We adopted a hierarchical labeling method for all the scans in the database to reduce human error. The radiologists involved with the project completed a stakeholder evaluation form. First, all 18,259 samples were obtained from the segmentation results generated by the proposed model and prelabeled using ITKSNAP software with masks marking the areas of the bone, aorta, carotid artery, and intracranial artery. They were multiclass classification with four classes. We annotated the arteries including the aorta, subclavian artery, common carotid artery, vertebral artery, internal carotid artery, basilar artery, posterior cerebral artery, middle cerebral artery, and anterior cerebral artery. Next, ten technologists with 2 or more years of postprocessing experience independently corrected the prelabeled images to obtain the preliminary labeling results. Then, two radiologists with 5 or more years of experience double-checked the accuracy of the preliminary labeling results and relabeled cases as needed to avoid mislabeling. When disagreements occurred between the two radiologists, an arbitration expert (a more senior radiologist with more than 10 years experience) made the final decision. Finally, we adopted the labeled source images that passed a quality inspection as the final labeling results. The average time required for each case is ~22 min. Thus, each CTA scan was annotated with four masks to obtain the reliable ground truth.

**Data preprocessing and augmentation.** Although the samples were labeled in a strict manner, scattered noise caused by operator misclicking cannot be entirely avoided; however, the effect was too small to determine through human inspection. To solve this problem, we used a connected component detection algorithm to eliminate scattered noise and improve the robustness of the data. In addition, data augmentation is a strategy to increase data diversity. Augmentation helps in training models more effectively and improves robustness properties. In this study, we augmented the training data by horizontal flipping, rotation with a maximum angle of 25°, shifting by horizontally and vertically translating the images (a maximum of twenty pixels), and random occlusion by selecting a rectangle region in an image and erasing its pixels with a random value. For a data set of size $N$, we generate a data set of 5 $N$ size. The rotation and shift were applied to improve the situation caused by the different body positions in the scanning process. The purpose of applying random occlusion was to avoid the artifacts caused by metal implants. Moreover, to improve model robustness, Gaussian noise was added to the training data.

**Model development.** This pipeline developed an automatic segmentation framework based on three cascaded ResU-Net models, which can remove the entire bone and fully extract the vessel. ResU-net is a modified U-net architecture with an added BR that can optimize the network and gain accuracy from a considerably increased depth[29,30]. ResU-Net2 was specially designed for bone edge optimization. Additionally, we proposed using CGPM to eliminate vascular segmentation errors and effectively avoid partial or missing vascular segments. The input data are integrated and continuous 3D spacing vascular images. The ResU-Net1 and ResU-Net2 models are responsible for the segmentation of the whole bone using a semantic segmentation under the same high resolution and for learning the physiological anatomical structural features that divide the vessel into the aorta, carotid artery, and intracranial artery based on the size of the artery. The ResU-Net3 model was adopted to achieve vascular segmentation according to the physiological anatomical structure. Finally, the proposed CGPM was applied to fix any ruptured vessels and ensure that they can feasibly be predicted and recognized at the morphological level (Fig. 9a, b). The architecture of CGPM is based on the 3D CNN and input data of the CGPM including the original CTA image without labeling, the current segmentation result generated by the combination of ResU-Net1, ResU-Net2, and ResU-Net3, and the labeled CTA images. The CGPM learned the location of the accurate partial missing and then accomplished the replenishment of the missed vessel. We also performed the ablation study to justify the contribution of each component in the pipeline (Fig. 9c).

**Training details.** We used BR as the principal architect in our network. Each layer in the BR is activated by the LeakyReLU function, which improves the flow of information and gradients throughout the network. BR can also identify the lowest loss value in the model due to its stack of three layers with $1 \times 1 \times 1$, $3 \times 3 \times 3$, and $1 \times 1 \times 1$ convolutions. In total, eight BRs are integrated into the subnetwork, each following four convolutional downsampling units and four convolutional upsampling units, respectively. This approach combines multiple network depths within a single network and represents complex functions at different scales. The feature volumes from each BR are first convolved to a smaller number of channels and then upsampled to the input size.

The input of the 3D CNN consisted of original CT slices with cropping and resolution normalization. First, we apply the layer thickness and space to normalize the CT volume size and crop it into patches with a dimension of $256 \times 256 \times 256$. These patches are the input data of the model, we combine the model output into the original CT volume. The 3D CNN was trained using the SGD optimizer with a momentum of 0.9, a peak learning rate of 0.1 for randomly initialized weights, a weight decay of 0.0001, and an initial learning rate of 0.01 that shrank by 0.99995 after each training step of 200,000 iterations. Each convolutional layer was activated by LeakyReLU (1) ($f(x) = 0.1^*x$ if $x < 0$, $f(x) = x$ if $x \geq 0$), and the sigmoid function (2) $\left( f(x) = \frac{e^x}{e^x + 1} \right)$ was used to predict the final segment probability results for each pixel. In this segmentation task, the loss function minimized $1 - D$ ($y_{\text{pred}}$, $y_{\text{target}}$), where $D$ is the Dice coefficient between the predicted and target segmentations. Thus, the Dice coefficient loss function is defined as follows: $y_{\text{target}}$ and $y_{\text{pred}}$ are the ground truth and binary predictions from the neural networks, respectively:

$$\text{Loss} = 1 - \frac{2 \left| y_{\text{pred}} \cap y_{\text{target}} \right|}{\left| y_{\text{pred}} \right| + \left| y_{\text{target}} \right|}. \tag{1}$$

After training the model with the training set, we used the validation set to validate the model and fine-tune the hyperparameters to optimize the model. All the experiments throughout this study were implemented based on Python and PyTorch libraries using a Linux server equipped with eight NVIDIA GeForce GTX 1080Ti GPUs with 11 GB of memory, and all the networks were trained from scratch using six of the GPUs.

For the failed cases in the validation process, vessel adhesion and vessel interruption were two main reasons for its occurrence. The vessel adhesion mostly occurred when the bone residual appeared around the vessel. Thus, the ResU-Net1 and ResU-Net2 were added to achieve bone segmentation. The vessel interruption often occurred where the blood flow was insufficient. This problem was solved by increasing the weight of the central region of the artery in the loss function, which enable the model to learn more characteristics about the central region. In addition, the CGPM, which targeted the occasional vessel interruption in the normal area, was proposed. When these failed cases occurred, more similar cases were added to train the model, which leads to better performance of the model.

**Auto postprocessing image layout.** The system achieves productivity and quality assurance characteristics by enabling traceability. Each image on the film can be traced and redirected to its original location in our image set. This system operates by separating the filming output process into two sub-tasks: verification and export. In the verification task, our program first processes skull subtraction and vessel extraction and generates four enlarged output images, including a VR, MIP, CPR, and MPR, indicating the position of the disease. Then, the four output images are placed in the first row of the film produced by the AI and can be verified and overwritten by technologists. After verification by the technologists, the reconstructed images can be exported to the PACS system.

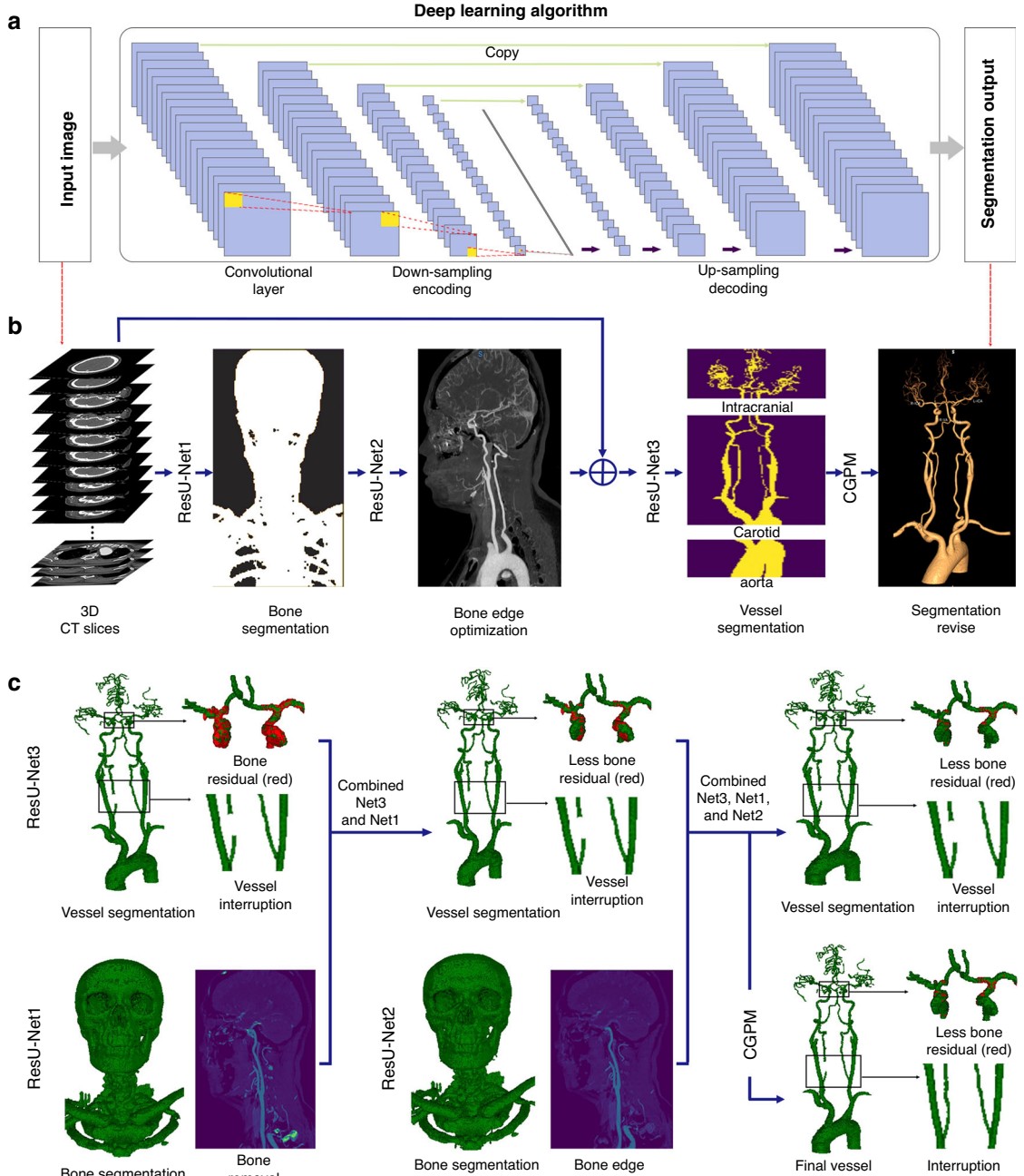

**Fig. 9 Bone and vessel segmentation framework of the CerebralDoc software. a** Schema and structure of the 3D ResU-net to automatically segment the skull and head and neck vessel from CTA slices. **b** The whole software pipeline is present. Skull segmentation and subtraction are automatically completed after two cascaded ResU-Nets firstly and the vessel is fully segmented by a physiological anatomical-based ResU-Net. After that, a revised model CGPM is proposed to fix any ruptured vessel by learning from the input data of the original CTA image without labeling, the current segmentation result generated by ResNet1, ResNet2, and ResNet3, and the labeled CTA images. **c** An ablation study was performed to justify the contribution from each component in the pipeline.

**Model effectiveness**. *DSC*: The performance of the proposed model was evaluated using DSC, which represents the overlap ratio between the ground truth and segmentation results and is defined as follows:

$$\mathrm{DSC}\left(Y,\widehat{Y}\right)=\frac{2\left|Y\cap\widehat{Y}\right|}{|Y|+\left|\widehat{Y}\right|}, \qquad (2)$$

where Y and $\widehat{Y}$, respectively, represent the ground truth and the binary predictions from the neural networks. Here, $\left|Y\cap\widehat{Y}\right|$ indicates the sum of each pixel value after calculating the dot product between the ground truth and the prediction. Two DSCs evaluated on both training and validation sets were used to avoid the question of excessively fits in the algorithm.

*V-score*: The *V*-score was innovatively proposed to reveal the continuity and integrity of vessel segmentation and is calculated according to the anatomy and morphological location of the vessel. A higher *V*-score value is correlated with a more critical vessel position; for example, the *V*-score of the proximal segments of a vessel is higher than that of its distal segments. The *V*-score represents the differences between the continuity of labels and predictions and is defined as follows:

$$V-\mathrm{score}=\frac{\sum\mathrm{weighted}\left(\widehat{Y}\cap Y\right)}{\sum\mathrm{weighted}(Y)}. \qquad (3)$$

*Recall*: Recall (also called sensitivity) is the proportion of positive cases that are correctly predicted as positive[31]. For a particular category, when a test image is an

input into a trained 3D-CNN model, the model outputs the probability that the image belongs to this category. A hard binary classification can be made by thresholding this probability $p \geq t$, where $t$ is a threshold value. Recall is defined in its various common appellations by the following equation:

$$\text{Recall} = \frac{\hat{Y} \cap Y}{Y}. \tag{4}$$

**Clinical evaluation**. The automatic head and neck CTA postprocessing system was designed to create a productive, quality-assured, and unattended image reconstruction process. The reconstruction images achieved by the technologists were regarded as comparable output and two experienced radiologists (10 years of experience) who were blinded to the processing conditions assessed whether the imaging quality fulfilled the clinical diagnostic requirements based on all processed image types (VR, MIP, CPR, and curved-MPR). All data sets were randomized and the qualification rates of image reconstruction were calculated separately. Subjective evaluations were scored as follows[32]: 5 = excellent image quality and contrast ratio, good vascular delineation, no artifacts, easy to diagnose; 4 = good image quality and contrast ratio, normal vascular delineation, with a few artifacts, but adequate for diagnosis; 3 = satisfactory image quality and contrast ratio, some artifacts, the artery was not clearly displayed but was sufficient for the diagnosis; 2 = weak image quality and contrast ratio, obvious artifacts, the artery was not clearly displayed and was not sufficient for diagnosis; 1 = poor image quality and contrast ratio, severe artifacts, difficult to distinguish the small artery and diagnosis was not possible. When the results differed between the 2 raters, a consensus was obtained.

We also compared the image quality, which was separately rated on VR, MIP, CPR, and curved-MPR based on the clinical request, which includes the vascular integrity, main cerebral vascular branches, vessel cleanness, wall calcification, degree of stenosis, and bone segmentation error. VR was graded mainly according to the vascular integrity, a branch of the main cerebral vascular and vessel cleanness as follows: 3 = good vascular delineation without interruption, excellent vascular side branch reconstruction, and clear vessel image; 2 = normal vascular delineations with partial interruption, good vascular side branch reconstruction with few residuals; 1 = whole vessel interruption, poor vascular side branch reconstruction and many residuals. MIP was graded based on the wall calcification and bone segmentation error as follows: 3 = no bone segmentation error and excellent vessel presentation; 2 = moderate bone residue but does not affect vessel observation; and 1 = severe bone residue and does affect vessel observation. CPR and curved-MPR were graded based on the vascular integrity and location and degree of stenosis, using the source images as "ground truth": 3 = good vascular delineation without interruption, the same degree of stenosis as the source image without omission; 2 = normal vascular delineation with partial interruption, and biased degree of stenosis compared to the source image without omission; 1 = whole vessel interruption, and severe narrow omission compared to the source image.

The clinical application value of CerebralDoc was assessed between July and November 2019 at Xuanwu Hospital, from the following aspects: 1. the coverage rate of all patients who underwent the head and neck CTA examination; 2. the overall, monthly, and daily postprocessing numbers, and the numbers successfully pushed to the PACS; 3. the average time required for postprocessing and the number of clicks by CerebralDoc and the technologists (for 100 randomly selected patients); 4. the clinical productivity over the tested months; and 5. the error rates of the model (the reasons were explored).

**Statistical analysis**. All the analyses were conducted using SPSS software version 23.0 (IBM, Armonk, New York). Categorical variables were expressed as frequencies (percentages, %), and continuous variables were expressed as the mean ± SD or as the interquartile range according to the normality of the data. Student's $t$-tests or Mann–Whitney $U$-tests were used to analyze the continuous data, and Fisher's test was used for the categorical data as appropriate. The interobserver reliability was assessed using kappa analysis.

**Reporting summary**. Further information on research design is available in the Nature Research Reporting Summary linked to this article.

## Data availability

Currently, the source imaging data from five tertiary hospitals cannot be made publicly accessible due to privacy protection. The CerebralDoc is accessible from the webserver created for this study at http://test.platform.shukun.net/login. User name: test, password: 123456. Request for software access and the remaining clinical evaluation images generated by technologists can be submitted to F.F. The source data underlying Supplementary Table 2, Figs. 3a, 7, and 8 are provided as a Source Data file with this paper.

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

## Acknowledgements
This work has been supported by "Beijing Natural Science Foundation (Z190014)" and "Beijing Municipal Administration of Hospital's Ascent Plan" (DFL20180802) and "Beijing Municipal science and technology projects" (Z201100005620009).

## Author contributions
The studies were conceptualized, results analyzed, and manuscript drafted by F.F., J.W. Y.X. and F.L. wrote the code, trained the models, and wrote the first draft of the manuscript, with guidance from J.L., M.Z., and D.R. Also, Y.L. and F.Y. provided imaging quality evaluation and data interpretation. C.Z. provided supervision of statistical analysis and interpretation of data. Y.S. provided case selection and annotations. Z.Y., Y.L, Y.C., and X.W. provided raw training data and overall study supervision. All authors were involved in critical revisions of the manuscript, and have read and approved the final version.

## Competing interests
The authors declare no competing interests.
