## [Peer Review File · Nature Communications]

Reviewers' Comments:

Reviewer #1:

Remarks to the Author:

In this study, the authors established an automatic post-processing of head and neck CTA to yield vessels and skull segmentation. They then assessed the performance of the model on an independent test set both quantitatively and qualitatively. The study is first of its kind with respect to the substantial dataset and the applied model, training and validation are flawless. However, the presented work suffers from major impediments.

Major comments:

1. The numerical performance assessment is not representative. While training and test set performance was measured using DSC, V-score and recall, the only measure that was reported for the test set was accuracy, which is in general a problematic measure for a highly unbalanced class distribution.
2. The labeling process and consequently the algorithm objective is not well defined in the manuscript. While the authors stated that the algorithm segments skull and vessels from CTA images, the annotation process described in the methods section only states annotation of "areas of the skull, aorta, carotid artery and intracranial artery". It is not clear if all vessels were annotated in the brain or only this part, were they all defined as one class or not. This brings in question the full objective during the training as well as the numerical performance assessment.
3. The clinical evaluation as was done in this study does not provide a valid measure. In order to provide a sensible qualitative evaluation, one must apply the assessment using blinded randomization, i.e. by comparison of the model's output to a comparable output when the reviewer is blinded to the source.

Minor comments:

4. The utilization of the term "postprocessing" in the manuscript title is quite ambiguous. I suggest to specify the exact operation (i.e. vessels segmentation)
5. The authors should provide descriptions of the occurrence of cerebrovascular diseases and specification in the dataset.
6. Since the results are presented before the methods section, all of the abbreviations must be introduced in the appropriate place (see for example page 6, BR, which is only introduced in the methods section in page 18). This included for example BR, CGPM, ResU-Net.
7. Page 21, DSC formulation, there seems to be a typo mistake in the denominator (').

Reviewer #2:

Remarks to the Author:

This paper presented a CTA postprocessing method based on common deep learning techniques. The main contribution of the work is the multi-institute, large-scale data and annotation involved in the experiment, and also the study involving real clinical application and evaluation. Method-wise the system is fairly standard, so there isn't much contribution along that front.

One major concern is that there is no comparison with any other methods, whether conventional or machine learning. Authors listed various previous methods, but none of them are evaluated in this study. Thus, there is no fact to support authors' claims that other methods have multiple weaknesses while the proposed method addressed them. Without such experiments, it is not

convincing to the readers that the method designed in this work is indeed proper for the task and better than other alternatives.

Apart from comparison, since there are 3 networks and an additional refinement step, ablation study for the contribution from each component used in the pipeline is needed to justify their values.

In addition, authors need to give examples for failed cases, what happened, and how they would be handled. It is not clear how CGPM "revise vascular segmentation errors and effectively avoid partial missing vasculature." It is also not clear from the text and figure what the purpose of the second network ResU-Net2 is.

Line 74, "image postprocessing" is a broad term, please list a few common ones.

Line 87-88, "they are either hand-crafted or insufficiently validated; thus, they have difficulty achieving the desired level of robustness", please give reference and/or examples to support this claim.

Line 100-103 3D-CNN is such a broad term that I am afraid this sentence does not hold much meaning without proper reference

Line 117 "Due to poor image quality, 507 scans were excluded", via manual inspection?

Line 125 and 127 full names of "BR" and "CGPM" appears way too late in the manuscript, so when they first appear, it is very confusing

Line 131-132 "The model training process worked perfectly, and all the curves reached convergence by the 20th epoch, reaching DSC of 0.956 and 0.914", "perfectly" is difficult to judge, it is better just to say "converge", and here two DSCs are not clear what the targets are.

Line 184 "was trained with optimal parameter selection by BRs", "optimal parameter selection" may not be an accurate description for BR, essentially all the networks are trained with data, so this "optimal" can be a very confusing and misleading term. This also applies to Line 359/375.

Line 334 "all 18,259 samples were pre-labeled using ITKSNAP software", entirely manual labelling or start from some segmentation results? Please also mention how long an average case takes for all 4 annotations. I think this number may not be the same as clinical processing time since they are different tasks.

Line 351 please provide more details of augmentation applied and though they seem just standard, maybe authors can give some perspective on why they are meaningful for the proposed task.

Last, the language is not satisfying, there are a lot of confusing expressions in the manuscript, it would be better to have some native speakers to fully revise the paper. To list some examples:

Line 93 "crucial information about arterial vessel status is not available", I have no idea what "crucial information" is.

Line 105 "based on an optimized physiological anatomically 3D-CNN", what is "physiological anatomically 3D-CNN"?

Line 217 "pipeline with guaranteed maximum patch-size", what do you mean by "guaranteed maximum"?

Line 352 what is "in multiples of five"??

Reviewer #3:

Remarks to the Author:

Dear Editor,

Thank you for the opportunity to review the manuscript "Rapid postprocessing of head and neck CTA using 3D convolutional neural network". I think that the manuscript is extremely interesting and that the number of dataset included ($n = 18766$) is huge.

The authors have created an intelligence postprocessing system supported by an optimized physiological anatomical-based 3D convolutional neural network that can automatically achieve CTA postprocessing in healthcare services

The authors showed that the system reduces the time consumed from 14.2 ± 3.6 min to 4.9 ± 0.4 min, the number of clicks from 115.87 ± 25.9 to 4 and the labor force from 3 to 1 technologist after five months application.

I think that the paper is very good and it could be accepted for publication.

Reviewer #4:

Remarks to the Author:

I am a practicing clinical neuroradiologist and evaluated this manuscript from the perspective of its potential clinical application. I do not have the expertise to evaluate the details of the data processing and model development except as they impact clinical workflow and diagnostic benefit.

With that in mind, my overall impression is that this is primarily an image processing descriptive technical manuscript. The evidence presented for its clinical application is "testimonial", based on acceptance ("doctor's trust" line 161) into the clinical workflow by the two reviewing radiologists and the presumed benefit of the streamlined processing steps (fewer clicks). No direct, objective comparison to images generated by other methods is presented and no data is presented specifically supporting the improved visualization of pathology in comparison to other methods (rule based, centerline tracking, active contour models and region growth methods on the same data as well as the proprietary processing software provided by the individual scanner vendors). This comparison is not included in the "quantitative" assessment (algorithmic performance, satisfaction of diagnostic needs and efficiency) mentioned in the introduction.

The authors also appear to have a bias that I have often seen from technical developers toward 3D surface rendered vessel images (the more "realistic" looking vessels) as they use these to present the pathology examples and errors encountered in their experience (Figure 3). In clinical practice, these 3D images are the least useful for diagnosis because the surface rendering frequently produces inaccurate depictions of the true vessel lumen by merging atherosclerotic calcification with similar density intravascular contrast material, obscuring intraluminal defects such as thrombus and by smoothing across small details such as small atherosclerotic plaque ulcerations or thin intimal dissection flaps.

The most widely practiced approach to interpretation of CTA images is to use multiplanar reformations (MPR) and/or thin-slab maximum intensity projections (MIP) for an overview of the vascular anatomy and then direct evaluation of the acquired source images in key areas such as the carotid bifurcation, cavernous sinus segments and vessel branch points for degree of stenosis, atherosclerotic irregularities or ulceration, bone segmentation errors, intimal dissection and aneurysm. Although the authors to present an example of MPR, CPR and MIP images in figure 7, it is not clear which of the processed image types were most relied upon to generate the 5-point ratings (see statistics Clinical evaluation section lines 148-153. Need comparison of specific pathology (above) visualization between this and other methods and also between the different outputs from this method, using the source images as "ground truth" (as pathology and DSA catheter angiography are unlikely to be available gold standards).

This same array of processed images should be presented with the examples in Figure 3 so that the reader can better evaluate the accuracy and possible errors of the method (it is likely that this will be favorable to the method with better visualization of the pathology and vessel lumen on MPR, CPR or MIP presentations).

On the positive side, the study demonstrates day-to-day clinical application of the method with a large number of cases and the ability to process images from a wide range of CT scanners and acquisition protocols. This is an important benefit for clinical application. The overall experience at the five imaging sites with reduction of time and the number of techs (3 to 1) in this specific hospital network is a positive, though anecdotal, demonstration.

Line-by-line comments, questions and minor corrections (underlined):

Line 49: ...proposed an artificial intelligence...

Line 54: how is "accuracy" defined? What was reference standard?

Lines 118-119: The age statistics presented here do not agree with Table 1.

Lines 137 and 138: 32 (1.75%) failed bone subtraction and 84 (5.15%) failed vessel segmentation.

Line 197: It is not clear how the post processing algorithm contributed to reduced radiation doses. The apparent reason for this statement is the acquisition of a large image volume, including both neck and head (see comments below).

Lines 224-233: The large volume of acquisition, including both head and neck is a compromise with optimal z-axis(slice thickness) resolution for the neck vessels but suboptimal for the smaller head vessels, which is why the general clinical practice is to image these two regions with two separate acquisitions, optimizing slice thickness and spatial resolution for each region. Dividing the total volume into 3 regions for post processing does not change or optimize the acquisition resolution (slice thickness and image matrix).

Line 253-254: The failure of the method with severe vessel stenosis is a major clinical application problem. This is the main reason clinical CTA studies are done, to identify and quantify clinically suspected stenosis or to confirm and quantify stenosis identified by ultrasound.

Line 258: Need to describe the correction "solved by the end of November".

Line 260: The misidentification of veins as arteries is a major problem with the AI method and data needs to be presented on the frequency and vessel characteristics associated with this error, including examples.

Line 268: It's not clear how the AI method can help the technologist remove artifacts by veins if there is a problem of misidentification of veins as arteries.

Line 274: It is not clear what "severely anomalous" origins of arteries means. Anomalous location such as a left vertebral arising directly from the aorta? Does anomalous mean severely stenotic origin?

Line 341: Who is the "arbitration expert"? A more senior radiologist?

Line 343: What is the "ground truth" used to annotate the four masks? The acquired source images?

FIGURES:

FIGURE 1 & 2: Validation of skull-vessel segmentation? Validated against what? Visual inspection of source images? Other, manufacturer post processing?

What is "intervascular rupture" in Figure 2? Based on 3D volume images only?

FIGURE 4: What is "qualification rate" ? "application efficiency" is compared to what?

FIGURE 5: Processing time comparison – is more data available compared to specific vendor processing software? Need more data by vessel (neck more labor intensive than head processing). Was diagnostic quality of AI processed images compared to manual processing in the visualization of pathology?

FIGURE 6 – appears to be emphasis on 3D surface rendered output. MPR, CPR and MIP also used?. How is "revised model CGPM" used to "fix" ruptured vessel? "Proposed" as a later feature of the method?

Response to Reviewer #1

Major comments:

1.The numerical performance assessment is not representative. While training and test set performance was measured using DSC, V-score and recall, the only measure that was reported for the test set was accuracy, which is in general a problematic measure for a highly unbalanced class distribution.

We thank you for raising this issue. In order to improve the performance assessment, we have incorporated the DSC, V-score and recall value of the testing process. The result showed that DSC of 0.951, V-scores of -0.936, and recall of 0.952 in the testing process. Please see “Result—Model performance” paragraph 1 lines 17-18 (page 7).

In this pipeline, we also did the bone segmentation to achieve the bone removal so that the artery segmentation is more accurate. The DSC and recall of bone segmentation have now been incorporated in the training and validation process. The accuracy we used in the testing process represents qualification rate of module, which reflects the performance in applications including both artery segmentation and bone segmentation. We also added this in the section of “Result—Model performance” paragraph 1 lines 13-15 (page 7) and Figure 2.

2.The labeling process and consequently the algorithm objective is not well defined in the manuscript. While the authors stated that the algorithm segments skull and vessels from CTA images, the annotation process described in the methods section only states annotation of “areas of the skull, aorta, carotid artery and intracranial artery”. It is not clear if all vessels were annotated in the brain or only this part, were they all defined as one class or not. This brings in question the full objective during the training as well as the numerical performance assessment.

We appreciate you for mentioning this issue. In order to further clarify the labeling process, more details of this process have been provided. The objective of the algorithm is to achieve both the bone segmentation and the artery segmentation. Bone

segmentation was done to achieve automatic bone removal and make the artery segmentation more accurate. Firstly, instead of annotating all vessels, we only labeled the arteries including aorta, subclavian artery, common carotid artery, vertebral artery, internal carotid artery, basilar artery, posterior cerebral artery, middle cerebral artery and anterior cerebral artery. In addition, the bones and arteries were classified as four classes: bone, aorta, carotid artery and intracranial artery, to improve the accuracy of the artery segmentation based on the physiological anatomy. The numerical performance assessment of vessels integrated aorta, carotid artery and intracranial artery. We have added this in the section of “Radiologist Annotations” paragraph 1 lines 4-8 (pages 17-18). In order to show it clearly, we also prepared a schematic diagram as shown below.

This figure indicates the labeling process in the source axial images. The intracranial artery (a), carotid artery (b) and aorta (c) were labeled in different color. The bone was also labeled to achieve bone removal and make the artery segmentation more accurate.

3.The clinical evaluation as was done in this study does not provide a valid measure. In order to provide a sensible qualitative evaluation, one must apply the assessment using blinded randomization, i.e. by comparison of the model’s output to a comparable output when the reviewer is blinded to the source.

Thanks for your suggestion. This is important for us to make the clinical evaluation more convincing. We have supplied the reconstruction images of the 152 patients

obtained manually using the proprietary processing software as a comparable output. After that, another two reviewers blinded to the two sources and gave scores separately. We compared the qualification rates of all reconstruction images, based on various diseases between AI processed images and manually processed images. We also compared the image quality which was rated on VR, MIP, CPR and curved MPR separately, based on the clinical requests including the vascular integrity, main cerebral vascular branches, vessel cleanness, wall calcification, degree of stenosis and bone segmentation error. The related part was revised in the section of “Clinical evaluation of CerebralDoc” paragraphs 1-3 (pages 7-8), “Discussion” paragraph 5 (pages 13-14), “Data preparation” paragraph 3 lines 3-5 (page 17) and “Clinical evaluation” paragraph 1 lines 2-8 and paragraph 2 (pages 23-25).

Minor comments:

4.The utilization of the term “postprocessing” in the manuscript title is quite ambiguous. I suggest to specify the exact operation (i.e. vessels segmentation).

Thanks for your advice. We have specified the term in the manuscript title. Based on the algorithm objective to achieve both bone and vessel segmentations, we revised the “postprocessing” into “vessel segmentation and reconstruction”.

5.The authors should provide descriptions of the occurrence of cerebrovascular diseases and specification in the dataset.

Thanks for your advice. Among the 18259 patients, 6300 patients were cerebrovascular diseases. The occurrence and specification of cerebrovascular diseases have been added in Table 1 of the dataset.

6.Since the results are presented before the methods section, all of the abbreviations must be introduced in the appropriate place (see for example page 6, BR, which is only introduced in the methods section in page 18). This included for example BR, CGPM, ResU-Net.

We appreciate you for pointing out this issue. In the revised manuscript, all of the abbreviations have been corrected in the right place (section of “Result-Model performance” paragraph 1 lines 2-5 page 6).

7. Page 21, DSC formulation, there seems to be a typo mistake in the denominator (‘).

Thanks for your suggestion. This point has been incorporated in the revised manuscript (section of “Model effectiveness” paragraph 1 line 4 page 22).

Response to Reviewer #2

1. One major concern is that there is no comparison with any other methods, whether conventional or machine learning. Authors listed various previous methods, but none of them are evaluated in this study. Thus, there is no fact to support authors’ claims that other methods have multiple weaknesses while the proposed method addressed them. Without such experiments, it is not convincing to the readers that the method designed in this work is indeed proper for the task and better than other alternatives.

We thank you for raising this issue. We compared the results generated by current method and that by the original method (only ResU-Net 3). The comparisons of DSC, V-score and Recall were 0.951 vs 0.925, -0.936 vs -0.916 and 0.952 vs 0.933, respectively. We also prepared a figure generated by algorithm to show it directly as follows. This result was added in the section of “Result—Model performance” lines 18-19 (page 7).

In this study, we focus on the artificial intelligence reconstruction system’s development, clinical evaluation of results and the efficiency of clinical application. This algorithm performed well using clinical datasets, and the distribution of BR and the proposed CGPM are creative. We checked our manuscript and found that the original expression in discussion part indeed needs improvement. In the revised

manuscript, we have rewritten “Discussion” paragraph 2 to reflect the fact that other methods in prior studies promote the development of purposed automated clinical vessel segmentation tool.

However, we totally agree that the comparison with other methods is valuable and we have read some recent relative researches and divided these methods into two categories, the rule-based methods and those using deep learning (DL), the Table below listed all the experimental results of the reported methods. Note that these methods were evaluated with different datasets using distinct evaluation metrics. Meanwhile, these studies may also use different datasets of different anatomic positions and different image modalities like CTA or MRA. The DL-based method proposed by Michelle Livne et al (2019) also used an optimized U-Net framework with high performance (a Dice value of ~ 0.88 , a 95HD of ~ 47 voxels and an AVD of ~ 0.448 voxels); however, this method only evaluated the cerebral vessel segmentation in MRA. It is rare to find the research that uses deep learning method to segment head and neck artery on CTA. In order to further verify the superiority of our framework, now we are trying to compare our model with precious DL-based framework using the same dataset. In this part of the work, we first need to reproduce previous DL-based algorithms and the consistent ratio of training, validation and testing sets and experimental settings are applied then. Sincerely hope you can give us some suggestions and this part of results will be shown in a future paper.

Table: Comparison with previous work, two categories of methods (rule-based and DL-based) are listed.

Type	Study	Test set patients	Image	AO	Carotid artery	Cerebral vessel
				DSC	DSC	DSC
Rule-based	Olivier Cuisenaire et al. (2009)	28	CTA	-	90.3	-
	Danilo Babin et al. (2013)	4	CTA	-	-	0.82
	Duan et al. (2016)	10	CTA	0.93	-	-

DL	Michelle Livne et al. (2019)	66	MRA	-	-	0.88
	Ours	1826	CTA	0.975		

Figure generated by the original version (only ResU-Net 3) of algorithm and the current version. In the original version, there were both vessel interruption and vessel adhesion. After the combination of ResU-Net 1, ResU-Net 2, ResU-Net 3 and CGMP, the problems of vessel interruption and vessel adhesion were solved and the performance was improved.

2. Apart from comparison, since there are 3 networks and an additional refinement step, ablation study for the contribution from each component used in the pipeline is needed to justify their values.

We appreciate you for this suggestion. In the revised paper, we have connected to our algorithm engineers and added the ablation study. The following figure indicated the contribution from each component to justify their values. For the vessel segmentation done by the ResU-Net 3, we can see the interruption and bone residual of vessels. The testing process reached dice similarity coefficient (DSC) of 0.925, weighted vessel score (V-scores) of -0.916 and recall of 0.933. The ResU-Net 1 was proposed to achieve the bone segmentation. Combining the ResU-Net 3 and ResU-Net 1, the bone

residual was improved and the indicators of DSC, V-scores and recall increased to 0.943, -0.929 and 0.948, respectively. The ResU-Net2 is responsible for optimization of the bone edge. By combining the ResU-Net 3, ResU-Net 1 and ResU-Net 2, the problem of bone residual was significantly reduced and the value of DSC, V-scores and recall raised to 0.947, -0.930 and 0.950, respectively. CGPM was proposed to eliminate vascular segmentation errors, so that partial or missing vascular segments were effectively avoided. We can see that the interruption of the vessel was corrected by adding the CGPM. The discussion above has been incorporated in Figure 9 and in the section of “Model Development” paragraph 1 last sentence (page 20).

	DSC	V-score	Recall
Only ResU-Net3	0.925	-0.916	0.933
ResU-Net3+ ResU-Net1	0.943	-0.929	0.948
ResU-Net3+ ResU-Net1+ ResU-Net2	0.947	-0.930	0.950
ResU-Net3 +ResU-Net1+ResU-Net2+CGPM	0.951	-0.936	0.952

3. In addition, authors need to give examples for failed cases, what happened, and how they would be handled. It is not clear how CGPM “revise vascular

segmentation errors and effectively avoid partial missing vasculature.” It is also not clear from the text and figure what the purpose of the second network ResU-Net2 is.

We thank you for proving the suggestion. The reasons for the failed case included vessel adhesion and vessel interruption. The vessel adhesion mostly happened when the bone remains around the vessel. In the revision, we added the bone segmentation to improve the performance of the vessel segmentation. As for the vessel interruption, it often happened where the blood flow was insufficient, which was addressed by increasing the weight of the central region of the artery in the loss function so that the model can learn more characters about the central region. In addition, we developed the CGPM which targeted on the occasional vessel interruption in the normal area. When these failed cases happened during the validation process, we adjusted the algorithm and added more similar cases to train the model, which leads to better performance of the model. The above discussion has been incorporated in the section of “Training details” Paragraph 4 (page 21) and figures for failed cases as follows.

In addition, we provided more details about CGPM to explain why this component could revise vascular segmentation errors and avoid partial missing vasculature. The architecture of CGPM is based on the 3D CNN proposed by Kamnitsas et al (K. Kamnitsas et al., Efficient multi-scale 3D CNN with fully connected CRF for accurate brain lesion segmentation, *Med. Image Anal.*, vol. 36, pp. 61–78, Feb. 2017). When vessel interruption happened, we collected the original CTA image without labeling, the current segmentation result generated by the combination of ResNet1, ResNet2 and ResNet3 and the labeled CTA images. These three parts were used as the input data of CGPM and it learned where the accurate partial missing location is. Then, the proposed CGPM could accomplish the replenishment of the missed vessel. We added this in the section of “Model Development—Network architecture” paragraph 1 line 15-29 (pages 19-20). We also prepared a schematic diagram as follows.

Regarding the purpose of ResU-Net 2, it was proposed to optimize the bone edge segmentation. We have added this in the section of “Model Development—Network

architecture” Paragraph 1 lines 4-5 (page 19). Figure 9 was revised to reflect the discussion as well.

This figure showed the failed cases during the process of model development. (a) The model misidentified bone as vessel which caused the bone residual. (b) A failed case showed the vessel adhesion. (c) The right carotid artery was not recognized which leads to the vessel interruption.

This figure showed the structure of the CGPM of a U-Net based architecture. The displayed U-net is an encoder-decoder network with a contracting path (encoding part, left side) that reduces the height and width of the input images and an expansive path (decoding part, right side) that recovers the original dimensions of the input images. Each box corresponds to a multi-channel feature map. The blank boxes stand for the

concatenated feature maps copied from the contractive path. The arrows stand for the different operations as listed in the right legend.

4.Line 74, “image postprocessing” is a broad term, please list a few common ones.

Thanks for your advice. Actually, the postprocessing of head and neck CTA means the reconstruction of the source image and generation of VR, MIP, CPR and curved-MPR. In the revised manuscript, “image postprocessing” was replaced by “image reconstruction” to make it more accurate.

5.Line 87-88, “they are either hand-crafted or insufficiently validated; thus, they have difficulty achieving the desired level of robustness”, please give reference and/or examples to support this claim.

Thanks for your advice. A reference was provided to support this statement in the section of “Introduction” paragraph 2 line 7 (page 4).

(Zhao FJ, Chen YR, Hou YQ, He XW. Segmentation of blood vessels using rule-based and machine-learning-based methods: a review. *Multimed Syst* 25,109-118)

6.Line 100-103 3D-CNN is such a broad term that I am afraid this sentence does not hold much meaning without proper reference

Thanks for your advice and proper references were given to support this claim in the section of “Introduction” paragraph 3 line 9 (page 5).

(Kamnitsas K. et al. Efficient multi-scale 3D CNN with fully connected CRF for accurate brain lesion segmentation. *Med Image Anal* 36, 61-78)

7.Line 117 “Due to poor image quality, 507 scans were excluded”, via manual inspection?

Thanks for your advice. Yes, the 507 scans were excluded via manual inspection including severe artifacts, delayed contrast agent, wrong scanned area, etc. This has been incorporated in the section of “Patients and image characteristics” paragraph 1 line 3 (page 6).

8.Line 125 and 127 full names of “BR” and “CGPM” appears way too late in the manuscript, so when they first appear, it is very confusing

We appreciate this issue you raised. In the revision, “BR” and “CGPM” were defined when they were used the first time in the section of “Result-Model performance” paragraph 1 lines 2-5 (page 6).

9.Line 131-132 “The model training process worked perfectly, and all the curves reached convergence by the 20th epoch, reaching DSC of 0.956 and 0.914”, “perfectly” is difficult to judge, it is better just to say “converge”, and here two DSCs are not clear what the targets are.

Thanks for your suggestion. We have deleted the word “perfectly” in the section of “Model performance” paragraph 1 line 10 (page 6) and just said “converge”. The target of DSC was to evaluate the segmentation performance of the model. Two DSCs were targeted on the training and validation sets respectively. Sometimes, the algorithm has problem of excessively fits when DSC is high on the training set and low on the validation set. Therefore, we need to evaluate the parameter on both training and validation sets. We added this in the section of “Model effectiveness” paragraph 1 lines 8-10 (page 24).

10.Line 184 “was trained with optimal parameter selection by BRs”, “optimal parameter selection” may not be an accurate description for BR, essentially all the networks are trained with data, so this “optimal” can be a very confusing and misleading term. This also applies to Line 359/375.

Thank you for raising this question. The function of BRs was to optimize the network and gain accuracy from considerably increased depth. In this study, we used the distribution of BRs in the network to find the lowest loss value in the model. So, we rephrased this sentence into “The model was developed with an optimized network by the distribution of BRs” in line 184 (page 10), “added BR that can optimize the network and gain accuracy from a considerably increased depth” in Line 359 (page 19) and “BR can also identify the lowest loss value in the model due to its stack of 3 layers with $1\times 1\times 1$, $3\times 3\times 3$ and $1\times 1\times 1$ convolutions” in line 375 (page 20).

11.Line 334 “all 18,259 samples were prelabeled using ITKSNAP software”, entirely manual labelling or start from some segmentation results? Please also mention how long an average case takes for all 4 annotations. I think this number may not be the same as clinical processing time since they are different tasks.

Thanks for your comment. These samples were started from some segmentation results generated by the proposed model. We added this in the section of “Radiologist Annotations” paragraph 1 lines 3-4 (page 17). An average case takes 22mins for all 4 annotations and it is longer than average clinical processing time. We added this in the section of “Radiologist Annotations” paragraph 1 lines 15-16 (page 18).

12.Line 351 please provide more details of augmentation applied and though they seems just standard, maybe authors can give some perspective on why they are meaningful for the proposed task.

We appreciate this suggestion. We augmented the training data by horizontal flipping, rotation with a maximum angle of 25 degrees, shifting by horizontally and vertically translating the images (a maximum of twenty pixels) and random occlusion by selecting a rectangle region in an image and erasing its pixels with a random value. The rotation and shift were applied to improve the situation caused by the different body positions in the scanning process. The purpose of applying random occlusion

was to avoid the artifacts caused by the metal implants. In the revision, these details of the data augmentation and the meaning of these augmentations have been provided in the section of “Data preprocessing and augmentation” lines 7-13 (pages 18-19).

13.Last, the language is not satisfying, there are a lot of confusing expressions in the manuscript, it would be better to have some native speakers to fully revise the paper.

We appreciate your suggestion about the language. We have deleted or corrected the confusing expressions mentioned below and revised the language of the paper by a native speaker called Dr. Iain C Bruce.

To list some examples:

Line 93 “crucial information about arterial vessel status is not available”, I have no idea what “crucial information” is.

We appreciate this suggestion. The expression of “crucial information” is ambiguous and we have rewritten this sentence into “accurate segmentation extraction of arterial vessel status is not available” in the line 93 (page 5).

Line 105 “based on an optimized physiological anatomically 3D-CNN”, what is “physiological anatomically 3D-CNN”?

“Physiological anatomically 3D-CNN” means a prior-knowledge based deep learning method, which aims to acquire information of three differentiation regions (aorta, carotid artery and cerebral artery) through bone segmentation information for better vessel segmentation. We revised it into “anatomy prior-knowledge based 3D-CNN” in the section of “Introduction” paragraph 3 line 2 (page 5).

Line 217 “pipeline with guaranteed maximum patch-size”, what do you mean by “guaranteed maximum”?

First, we use layer thickness and layer spacing to normalize the CT volume size, then crop it into patches with the size of $256*256*256$. These patches are the input data of the model, then we combine the model output into the original CT volume. We added this in the section of “Training details” paragraph 2 lines 2-4 (page 20).

Line 352 what is “in multiples of five”??

We have revised this expression into “For a dataset of size N, we generate a dataset of 5N size” in the revised manuscript section of “Data preprocessing and augmentation” lines 10-11 (page 18).

Response to Reviewer #3:

I think that the manuscript is extremely interesting and that the number of dataset included ($n = 18766$) is huge.

The authors have created an intelligence postprocessing system supported by an optimized physiological anatomical-based 3D convolutional neural network that can automatically achieve CTA postprocessing in healthcare services

The authors showed that the system reduces the time consumed from 14.2 ± 3.6 min to 4.9 ± 0.4 min, the number of clicks from 115.87 ± 25.9 to 4 and the labor force from 3 to 1 technologist after five months application.

I think that the paper is very good and it could be accepted for publication.

Thanks for your recognition of our research sincerely.

Response to Reviewer #4

1. With that in mind, my overall impression is that this is primarily an image processing descriptive technical manuscript. The evidence presented for its clinical application is “testimonial”, based on acceptance (“doctor’s trust” line

161) into the clinical workflow by the two reviewing radiologists and the presumed benefit of the streamlined processing steps (fewer clicks). No direct, objective comparison to images generated by other methods is presented and no data is presented specifically supporting the improved visualization of pathology in comparison to other methods (rule based, centerline tracking, active contour models and region growth methods on the same data as well as the proprietary processing software provided by the individual scanner vendors). This comparison is not included in the “quantitative” assessment (algorithmic performance, satisfaction of diagnostic needs and efficiency) mentioned in the introduction.

We thank you for this suggestion and agree that the quantitative comparison between CerebralDoc and other methods is crucial. So, in the section of “Clinical evaluation”, we supplied the same images of 152 patients produced by technologists who worked on the proprietary processing software (AW4.7 GE Healthcare) including the centerline tracking and the region growth method.

In order to provide a sensible qualitative evaluation, we apply the assessment using blinded randomization in this section. By comparing the model’s output to a comparable output generated by the technologists on the workstation, in which the reviewers are blinded to the source and give the score. Also, we evaluated the differences of the score between AI processed image and manual processing based on various diseases including atherosclerosis, cerebrovascular disease, arterial aneurysm, and vascular variation. The results have been added in the section of “Clinical evaluation of CerebralDoc” paragraph 1-3 (pages 7-8).

As for the improved visualization of pathology, we also compared different output of this method to human output rated on VR, MIP, CPR and curved MPR separately based on the clinical requests including the vascular integrity, main cerebral vascular branches, vessel cleanness, wall calcification, degree of stenosis and bone segmentation error. We found that there were statistical differences with respect to VR ($P<0.001$) and MIP ($P<0.001$) in favor of CerebralDoc. All results were presented

in the section of “Clinical evaluation of CerebralDoc” paragraphs 1-2 (pages 7-8) and the images were added to show it directly. Also, we changed the discussion in the section of “Discussion” paragraph 5 (pages 13-14) and added a description of method in the section of “Clinical evaluation” paragraph 1-2 (pages 23-25).

2. The authors also appear to have a bias that I have often seen from technical developers toward 3D surface rendered vessel images (the more “realistic” looking vessels) as they use these to present the pathology examples and errors encountered in their experience (Figure 3). In clinical practice, these 3D images are the least useful for diagnosis because the surface rendering frequently produces inaccurate depictions of the true vessel lumen by merging atherosclerotic calcification with similar density intravascular contrast material, obscuring intraluminal defects such as thrombus and by smoothing across small details such as small atherosclerotic plaque ulcerations or thin intimal dissection flaps. The most widely practiced approach to interpretation of CTA images is to use multiplanar reformations (MPR) and/or thin-slab maximum intensity projections (MIP) for an overview of the vascular anatomy and then direct evaluation of the acquired source images in key areas such as the carotid bifurcation, cavernous sinus segments and vessel branch points for degree of stenosis, atherosclerotic irregularities or ulceration, bone segmentation errors, intimal dissection and aneurysm. Although the authors to present an example of MPR, CPR and MIP images in figure 7, it is not clear which of the processed image types were most relied upon to generate the 5-point ratings (see statistics Clinical evaluation section lines 148-153. Need comparison of specific pathology (above) visualization between this and other methods and also between the different outputs from this method, using the source images as “ground truth” (as pathology and DSA catheter angiography are unlikely to be available gold standards). This same array of processed images should be presented with the examples in Figure 3 so that the reader can better evaluate the accuracy and

possible errors of the method (it is likely that this will be favorable to the method with better visualization of the pathology and vessel lumen on MPR, CPR or MIP presentations).

Thanks for your suggestion. Sorry for the bias in prior Figure 3 and your advice is of much value. We only used VR to represent the postprocessed images of several disease which is improper in clinical practice. Thus, the same arrays of processed images in original Figure 3 were supplied in the revised Figure 5 (a, b, c) and Figure 6 so that the reader can better evaluate the accuracy and possible errors of the method.

The original Figure 7 was used to show the output images of CerebralDoc. In order to present the comparison of specific pathology (above) visualization between this method and other methods and also between the different outputs from this method, we added these content in Figures 3-6. After evaluating different output of this method to human output rated on VR, MIP, CPR and curved MPR separately, we found that CerebralDoc had better visualization of the pathology in VR ($P < 0.001$) and MIP ($P < 0.001$). We presented this in Figure 3B and Figure 4. Besides, the VR, MIP, CPR and curved MPR were considered together to generate the 5-point ratings which reflected whether the image quality fulfills the clinical diagnostic requests. We added this in the section of “Clinical evaluation” paragraph 1 lines 6-7 (page 24).

On the positive side, the study demonstrates day-to-day clinical application of the method with a large number of cases and the ability to process images from a wide range of CT scanners and acquisition protocols. This is an important benefit for clinical application. The overall experience at the five imaging sites with reduction of time and the number of techs (3 to 1) in this specific hospital network is a positive, though anecdotal, demonstration.

Line-by-line comments, questions and minor corrections (underlined):

3.Line 49: ...proposed an artificial intelligence...

Thanks for your suggestion. We have revised this sentence in the Line 49 (page 3).

4.Line 54: how is “accuracy” defined? What was reference standard?

Thanks for your question. The “accuracy” of independent testing dataset in line 54 is the ratio between the number of correctly segmented images and the total number of testing images. The reference standard was found as follows.

(Hao Xiong, Peiliang Lin, Jingang Yu, et al. Computer-aided diagnosis of laryngeal cancer via deep learning based on laryngoscopic images. *EbioMedicine*. 2019) and (Zhao F, Chen Y, Chen F, et al. Semi-supervised cerebrovascular segmentation by hierarchical convolutional neural network. *IEEE Access*. 2018).

5.Lines 118-119: The age statistics presented here do not agree with Table 1.

We apologize for the mistake. In the revised manuscript, the age statistics have been corrected in the section of “Result” paragraph 1 line 3 (page 6).

6.Lines 137 and 138: 32 (1.75%) failed bone subtraction and 84 (5.15%) failed vessel segmentation.

Thanks for your suggestion. This sentence has been corrected in the line 137 (page 7).

7.Line 197: It is not clear how the post processing algorithm contributed to reduced radiation doses. The apparent reason for this statement is the acquisition of a large image volume, including both neck and head (see comments below).

8.Lines 224-233: The large volume of acquisition, including both head and neck is a compromise with optimal z-axis(slice thickness) resolution for the neck vessels but suboptimal for the smaller head vessels, which is why the general clinical practice is to images these two regions with two separate acquisitions, optimizing slice thickness and spatial resolution for each region. Dividing the total volume into 3 regions for post processing does not change or optimize the acquisition resolution (slice thickness and image matrix).

Thanks for your question. We actually acquired both neck and head at the same time using a large image volume. In order to optimize the acquisition resolution, we used the 3D smart mA to ensure to image quality and the radiation dose was 2.25-2.49mSv. Dividing the total volume into 3 regions is a part of the model development, instead of the postprocessing. The purpose of dividing 3 regions was to achieve better vessel segmentation and extraction of the model since the sizes of the vessels were different in aorta, carotid and intracranial.

The post processing algorithm contributed to reduced radiation doses by automatic bone removing for head and neck CTA reconstruction instead of relying on subtraction by two scans (both plain scan and enhanced scan). The algorithm only needs the enhanced scan to achieve the bone extraction and finish the post-processing, resulting in a lower patient radiation dose (the plain scan not needed). We have explained this in the section of “Discussion” Paragraph 5 lines 18-19 (pages 13-14).

9.Line 253-254: The failure of the method with severe vessel stenosis is a major clinical application problem. This is the main reason clinical CTA studies are done, to identify and quantify clinically suspected stenosis or to confirm and quantify stenosis identified by ultrasound.

Thanks for your question. The failure of severe vessel stenosis cases happens during the process of independent clinical evaluation when the model was first applied in the clinical practice. At that time, we can see the incidence of the vascular interruption caused by severe vessel stenosis failure in this method is very low (3.9%, 6/152) and most of them showed partial vessel interruption. The figure below shows the difference between CerebralDoc and human manual reconstruction images.

Since this is the main reason clinical CTA studies are needed, the algorithm engineers tried to solve this problem although it happened rarely. After collecting these failed cases happened from the clinical applications, these cases would be labeled to train the CerebralDoc and solve this problem. We added this part in the section of “Discussion” paragraph 5 lines 23-24 (page 14).

a. The image generated by AI showed a vessel interruption in the left internal carotid. b. In the image generated by manual processing, this vessel is continuous with luminal stenosis. c. A partial interruption can be seen in the MIP generated by AI while it was normal in the manually processed image (d).

10.Line 258: Need to describe the correction “solved by the end of November”.

Thanks for your advice. The clinical radiologists labeled these failed cases as unqualified cases in the CerebralDoc and these cases would be recollected by algorithm engineers. These cases were relabeled and trained the CerebralDoc, so this problem has been mostly solved at the end of the November. We added this part in the section of “Discussion” paragraph 5 lines 23-24 (page 14).

11.Line 260: The misidentification of veins as arteries is a major problem with the AI method and data needs to be presented on the frequency and vessel characteristics associated with this error, including examples.

Thanks for your advice. The frequency of the misidentification of vein as arteries was 1.3% (2/152) in the section of “Clinical evaluation of CerebralDoc” paragraph 3 line 4

(page 8). It happened when the contrast phase severely delayed, and the vein CT value was the same as the artery. We added this in the section of “Discussion” paragraph 5 lines 24-25 (page 14). An example of this case was added as follows and in Figure 5 of this manuscript.

This figure showed the CerebralDoc misidentified the jugular vein as carotid artery.

12.Line 268: It’s not clear how the AI method can help the technologist remove artifacts by veins if there is a problem of misidentification of veins as arteries.

Thanks for your question. The AI misidentified veins as arteries only happened when the contrast phase severely delayed, and the vein CT value was the same as the artery. In these cases, we also can see the vein artifacts to some degree from the human output (we added the figure of manual processing of the same case showed above). In most of the cases, AI would not have this problem and thus can help the technologist remove artifacts of veins. We specified this point in the section of “Discussion” paragraph 6 lines 6-8 (page 14).

This figure showed the severe vein artifacts in the manually processed image of VR, MIP and CPR.

13.Line 274: It is not clear what “severely anomalous” origins of arteries means. Anomalous location such as a left vertebral arising directly from the aorta? Does anomalous mean severely stenotic origin?

Thanks for your question. The left vertebral arises directly from the aorta and severely stenotic origins were relatively common in the clinical practice, thus the CerebralDoc have already reconstructed this kind of images. We showed examples generated by AI in the following figure. Actually, the “severely anomalous” origins means the abnormal vessel routine which is relatively rare in the clinical scenarios. We have added this in the section of “Discussion” paragraph 7 lines 2-4 (pages 14-15).

This figure showed the postprocessing images of the CerebralDoc. The first line showed the severely stenotic origin of left vertebral artery and the second line showed a left vertebral arising directly from the aorta.

14.Line 341: Who is the “arbitration expert”? A more senior radiologist?

Yes, the arbitration expert means a more senior radiologist (with more than 10 years experiences). We added this in the section of “Radiologist annotations” line 13 (page 20).

15.Line 343: What is the “ground truth” used to annotate the four masks? The acquired source images?

Thanks for your question. Yes, the acquired source images were used as the “ground truth” and we annotated the four masks at the source images. We supplied this in the section of “Radiologist annotations” lines 13-14 (page 18).

FIGURES:

16.FIGURE 1: Validation of skull-vessel segmentation? Validated against what? Visual inspection of source images? Other, manufacturer post processing?

Thanks for your question. The validation of bone and vessel segmentation was used to make sure the algorithm indicators acceptable in training process. This is the architecture of algorithm in which we divide all cases into two parts: one for training the model and the other for validating the algorithm indicators. They were validated against the annotated source images rather than the visual inspection of source images or the manufacturer post processing. We added this explanation in the Figure legends of Figure 1 (page 31).

17.What is “intervascular rupture” in Figure 2? Based on 3D volume images only?

Thanks for your question. We have changed the “intervascular rupture” into “vascular interruption” in Figure 2 to make it more accurate. We evaluated the vascular interruption in the post-processing image based on 3D volume image mainly, because this sequence could reflect it directly. After that, we also checked the source image to avoid the vessel occlusion.

18.FIGURE 4: What is “qualification rate”? “application efficiency” is compared to what?

The qualification rate = AI successfully processing numbers/ Total clinical CTA numbers. We added this in original Figure 4 (current Figure 7, page 32). Additionally, we want to show the clinical application of the CerebralDoc from July to November 2019. Therefore, there is no comparison and we corrected the expression in the Figure legend of Figure 7 (page 32).

19.FIGURE 5: Processing time comparison – is more data available compared to specific vendor processing software? Need more data by vessel (neck more labor intensive than head processing). Was diagnostic quality of AI processed images compared to manual processing in the visualization of pathology?

Thanks for your suggestion. We acquired these two regions at the same time without separation. In order to optimize the spatial resolution, we used the 3D smart mA to achieve required image quality and to make sure the radiation dose was 2.25-2.49mSv. The slice thickness was 0.625mm for the reconstruction and we uploaded the 2mm slice thickness to the PACs. The postprocessing of the head and neck was done at the same time, so the separated time of head and neck was difficult to obtain. We hope that you could understand it and if you have any further questions, please contact us.

We agree that the comparison of diagnostic quality between AI processed images and manual processing is of great significance. However, the diagnostic quality is not just depends on the processed images in clinical practice. The radiologists make a diagnosis based on both the processed image and the source image, so even though the visualization of pathology was improved in AI processed image, the evaluation of diagnostic quality may not show any difference due to the effects of the source images. The advantages of CerebralDoc in clinical practice was labor and time-saving. However, this point is of great value and we will do this in the further study when AI could do the diagnosis. We will compare the diagnostic quality between AI on CerebralDoc and radiologists on manual processing.

20.FIGURE 6 – appears to be emphasis on 3D surface rendered output. MPR, CPR and MIP also used? How is “revised model CGPM” used to “fix” ruptured vessel? “Proposed” as a later feature of the method?

Thanks for your question. Yes, MPR, CPR and MIP were also used. So, we have replaced one of the images to MIP in original Figure 6 (current Figure 9). We mainly used VR to show the process of the algorithm, because it looks more intuitive and could reflect the segmentation performance of algorithm directly.

CGPM was used to fix the ruptured vessel mainly by the typical structure called U-Net based architecture. The input data of CGPM included the original CTA image without labeling, the current segmentation result generated by combining ResU-Net1, ResU-Net2 and ResU-Net3 and the labeled CTA images. From these images, it learned where the accurate partial missing location is and then accomplish the replenishment of the missed vessel. We added this in the section of “Model Development” paragraph 1 lines 15-19 (pages 19-20) and in the Figure legends of original Figure 6 (current Figure 9, page 33).

Moreover, the proposed CGPM was a part of the model instead of a later feature which targeted the occasional vessel interruption in the normal area. We added this sentence in the section of “Training details” paragraph 4 lines 7-8 (page 21).

Reviewers' Comments:

Reviewer #1:

Remarks to the Author:

The authors have applied extensive work yielding tremendous improvement to the manuscript. I am happy to approve the manuscript with only few minor comments to address:

1. In the methods section under Data preparation, the authors describe the datasets splits into "training process" and "testing process". Although the authors added according to our suggestions additional performance measure for the test set, it is still only described as "used for an accuracy assessment", please update it to the added performance-measures or use a more general term "performance assessment".
2. I thank the authors for the detailed answer regarding the labelling process. The authors have significantly improved the description under the "Radiologist annotations" section. To unequivocally finalize it, please add the total number of labelled classes to the paragraph (i.e. binary classification / multiclass classification with X classes).
3. The methods-comparison table that the authors added in response to reviewer #2 comment #1 is quite informative and provides an important scientific context to the presented work. I therefore suggest adding it to the supplementary material.

Reviewer #2:

Remarks to the Author:

Authors made a thorough revision and addressed my questions.

Reviewer #4:

Remarks to the Author:

This is the second review of this manuscript following the authors' response to reviewer comments and questions. The revised manuscript addresses my comments and questions thoroughly with clarification and expansion of the clinical evaluation and substantial improvement in the presentation of representative and instructive images in the figures. Based on these improvements, I can recommend this manuscript for publication.

Response to Reviewer #1:

The authors have applied extensive work yielding tremendous improvement to the manuscript. I am happy to approve the manuscript with only few minor comments to address:

1. In the methods section under Data preparation, the authors describe the datasets splits into “training process” and “testing process”. Although the authors added according to our suggestions additional performance measure for the test set, it is still only described as “used for an accuracy assessment”, please update it to the added performance-measures or use a more general term “performance assessment”.

Thanks for your suggestion. We have changed the “used for an accuracy assessment an accuracy” into “used for performance assessment” in the section of “Methods” paragraph 2 line 7 (page 17).

2. I thank the authors for the detailed answer regarding the labelling process. The authors have significantly improved the description under the “Radiologist annotations” section. To unequivocally finalize it, please add the total number of labelled classes to the paragraph (i.e. binary classification / multiclass classification with X classes).

Thanks for your advice. We have specified this sentence into “They were multiclass classification with four classes” in the section of “Radiologist annotations” paragraph 1 lines 5-6 (page 18).

3. The methods-comparison table that the authors added in response to reviewer #2 comment #1 is quite informative and provides an important scientific context to the presented work. I therefore suggest adding it to the supplementary material.

Thanks for your comments. We are glad to add this to the supplementary material as Supplementary Table 3. We supplied this in the section of “Discussion” paragraph 2 line 4 (page 10).

Response to Reviewer #2:

1. Authors made a thorough revision and addressed my questions.

Thanks for your comments of our research sincerely.

Response to Reviewer #4:

1. This is the second review of this manuscript following the authors' response to reviewer comments and questions. The revised manuscript addresses my comments and questions thoroughly with clarification and expansion of the clinical evaluation and substantial improvement in the presentation of representative and instructive images in the figures. Based on these improvements, I can recommend this manuscript for publication.

We appreciate your comments and thank you very much.